# The footprint of column collapse regimes on pyroclastic flow temperatures and plume heights

Matteo Trolese [1], Matteo Cerminara [2], Tomaso Esposti Ongaro[2] & Guido Giordano[1]

The gravitational collapse of eruption columns generates ground-hugging pyroclastic density currents (PDCs) with highly variable temperatures, high enough to be a threat for communities surrounding volcanoes. The reasons for such great temperature variability are debated in terms of eruptive versus transport and emplacement processes. Here, using a three-dimensional multiphase model, we show that the initial temperature of PDCs linearly correlates to the percentage of collapsing mass, with a maximum temperature decrease of 45% in the case of low percentages of collapse (10%), owing to an efficient entrainment of air into the jet structure. Analyses also demonstrate that column collapse limits the dispersal capabilities of volcanic plumes, reducing their maximum height by up to 45%. Our findings provide quantitative insights into the mechanism of turbulent mixing, and suggest that temperatures of PDC deposits may serve as a marker for determining column collapse conditions, which are of primarily importance in hazard studies.

[1] Dipartimento di Scienze, Sezione di Geologia, Università degli Studi Roma Tre, L.go S. Leonardo Murialdo 1, 00146 Roma, Italy. [2] Istituto Nazionale di Geofisica e Vulcanologia, Sezione di Pisa, Via della Faggiola 32, 56126 Pisa, Italy. Correspondence and requests for materials should be addressed to M.T. (email: matteo.trolese@uniroma3.it)

In explosive eruption columns, the heat exchange between the mixture of gas and fragmented magma, and the entrained atmosphere controls the plume dynamics. When this exchange is efficient enough, columns become buoyant, and the most powerful ones are able to reach the stratosphere, dispersing great volumes of material on a global scale, and perturbing the climate system[1–3]. In contrast, when the eruptive jet fails to mix with sufficient ambient air to become buoyant, the eruption column collapses—totally or partially—and generates hot pyroclastic density currents (PDCs) that propagate laterally on the ground, threatening the people surrounding volcanoes[4,5]. There is a continuum of collapsing regimes between these end-members, where the two phases can coexist simultaneously in a single eruption.

The study of the source conditions that promote the partial or total collapse of the erupting mixture has attracted attention[6–12], owing to its fundamental role for understanding the dynamics of eruption column formation and associated PDCs. Constraining how much of the erupted mass is partitioned between the buoyant and collapsing regions is critical to identifying all sources of volcanic hazards, as this determines both the amount of material dispersed in the atmosphere and the amount deposited by PDCs. To date, the mass eruption rate and the initial velocity at the vent (with the additional effect of the crater shape, which drives the decompression of the mixture[13,14]) are considered to be the most important factors in determining the destabilization of the column[11,15,16]. Conditions at collapse, in turn, are directly translated into spatial and temporal variations in PDC feeding conditions. However, until recently, there have been no quantitative studies investigating whether and how different collapse regimes relate to different thermal conditions of the mass entering the PDC. An understanding of the mechanisms exchanging magmatic heat with the atmosphere during the formation of PDCs is fundamental to explain the great variety of their thermal impacts on invasion areas[5,17], as well as the great variety of their emplacement temperatures[18–21].

With the growing availability of computational power, modeling studies have begun to explore the three-dimensional (3D) dynamics of turbulent volcanic plumes[14,16,22–26]. It has recently been demonstrated that 3D numerical models can reproduce, with quantifiable uncertainty, the dynamics of the turbulent mixing between the ascending mixture and the atmosphere forming convective plumes[23,25,27]. Unstable regimes eventually leading to partial collapses have also been recognized[10,11,23], which can explain several depositional features observed in the field[28] and characterized in the laboratory[12,29].

Here, we use an existing 3D flow model[23,24] that accounts for the main physical regimes of volcanic plumes to quantify, for the first time, collapsing regimes both in terms of the relative amount and mean temperature of the mass involved in the collapse phase. We show that depending on the amount of mass being partitioned between the convective and collapsing regions of the column, the total plume height and the initial temperature of PDCs can be substantially reduced. Collectively, our results reveal the importance of accounting for column collapse regimes in order to fully assess the dispersal of the pyroclasts in the atmosphere and the severity and spatial distribution of PDC impacts on areas inundated downstream.

## Results

**End-member scenarios.** We employed a 3D model that simulates the large-scale features of polydisperse gas–particle mixtures, including preferential concentration (i.e., nonhomogeneous particle clustering in turbulence) and gravitational settling of particles. The model has been extensively tested against numerical and experimental benchmarks[25,27]. Turbulence is modeled via the dynamic Large Eddy Simulations approach, enabling efficient and accurate turbulent spectrum calculations without the need of any empirical parameter. Boundary conditions represent the stationary injection of an initially homogeneous mixture of gas (assumed to be water vapor) and pyroclasts (grain size in the ash range) from a circular crater above a topographically flat surface into a stratified atmosphere (meteorological conditions are shown in Supplementary Fig. 1). Although strong wind conditions may in principle enhance the entrainment rate, we chose to use a windless atmosphere for all simulations, as we did not focus on quantifying the effect of wind on the plume dynamics. We then conducted a series of 3D experiments exploring different source conditions (Supplementary Table 1), which are determined by imposing the flow parameters at the conduit exit, such as pressure, mixture gas content, temperature, velocity equal to the speed of sound (choked-flow constraint), and mass eruption rate. The eruption source parameters of our model are consistent with those of typical explosive Plinian eruptions[30–32]. For overpressured flows, conditions after isentropic decompression are computed following ref. [33] and assuming a correctly expanded[13], homogeneous (i.e., with a top-hat density and velocity profile) jet at atmospheric pressure. We thus disregard the effects of potential under- or over-expansion of the mixture in the volcanic crater[14,34,35]. Recent numerical studies[14,36] have demonstrated that compression/decompression patterns, and the presence of shocks and tangential discontinuities are able to promote collapse conditions due to strong density and velocity inhomogeneity. Such phenomena are not considered in this paper. To examine the effects of gas–particle kinematic decoupling on the dynamics of turbulent plumes, all simulations are run twice, once with particles of two sizes, which represent the coarse (500 μm) and fine ash (50 μm) in equal proportions, and once with only a fully coupled particle class (i.e., dusty-gas model[37]). A detailed description of the numerical model, modeling procedure and source conditions assumed for the simulated scenarios are given in the Methods section.

Our 3D experiments explore the full spectrum of volcanic column regimes, from fully convective plumes to total collapsing fountains generating PDCs, in a mass eruption rate range spanning two orders of magnitude (from $10^7$ to $10^9$ kg s$^{-1}$). For each simulation, we determine the percentage of collapsing mass $Q_c$ and the average temperature of the mixture in the collapsing region $T_c$ (as detailed in the Methods section), and highlight the role of turbulence and mixing of atmospheric air in decreasing the temperature of the collapsing mass for models in which gas and particles are either in equilibrium or in kinematic decoupling regime.

Partial and total collapse regimes with gas–particle kinematic decoupling are shown in Fig. 1 for a mass eruption rate of $10^8$ kg s$^{-1}$ (input parameters are reported in Table 1). In the first case (Fig. 1a, b), the column is characterized by a regime of incipient collapse, in which batches of mass are intermittently released throughout the simulation time. At about 1 km above the inlet, the inner and hotter portion of the column starts displaying oscillatory behavior, driven by the development of turbulent large eddies along the margins. This turbulent mixing progressively erodes the central unmixed laminar jet, favoring the entrainment of cold atmospheric air and the subsequent dilution of the mixture (Fig. 1a, Supplementary Fig. 2a, b). The degree and extent of this process determine the percentage of mass involved in the collapse (i.e., the stronger the erosion, the less the collapsing mass). Given that the erosion process does not involve the entire central structure before the upward momentum of the mixture is exhausted, the less mixed, dense part of the column collapses toward the ground, in this case from a height of ~2.8 km (collapse height—$H_c$; Methods section). During the collapse, the mixture continues entraining

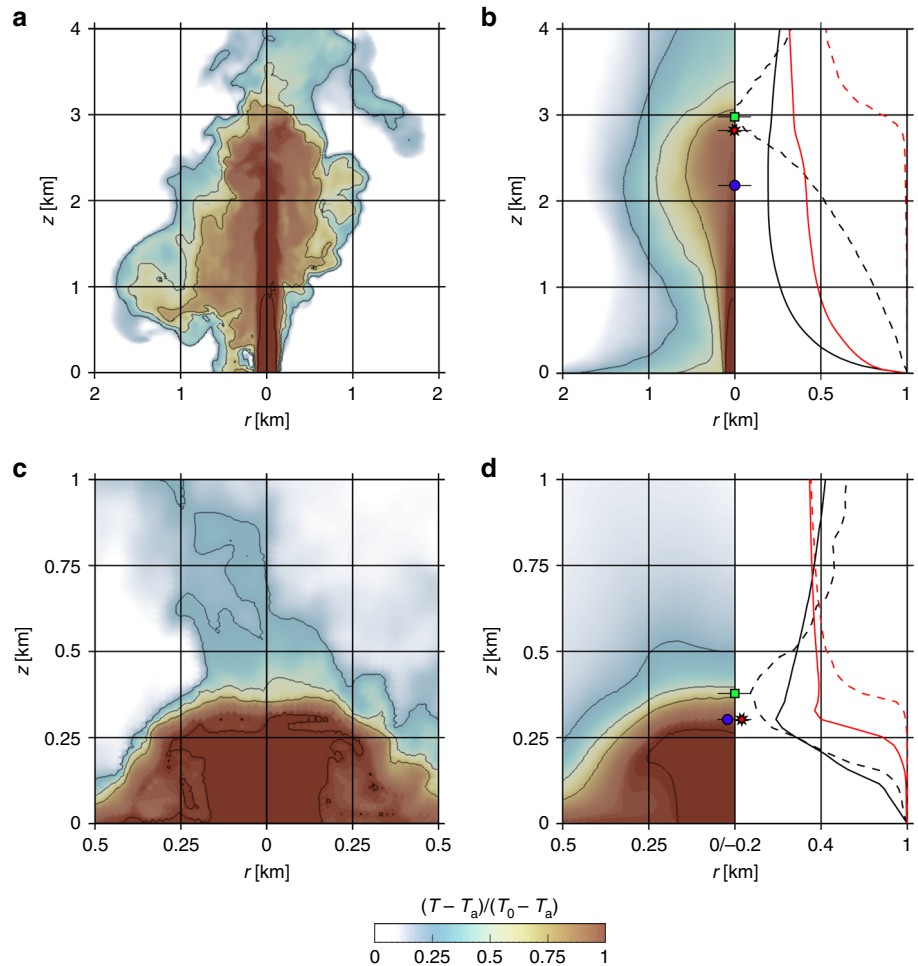

**Fig. 1** Comparison between the internal structures of the two column collapse end-members. Input conditions are reported in Table 1. **a**, **c** Cross-sectional images showing the instantaneous distribution of the temperature $T$ difference relative to the stratified atmospheric temperature $T_a$ at the same vertical position normalized to the initial temperature $T_0$, at 100 and 315 s respectively, for the partial (**a**) and total (**c**) collapse. Isolines correspond to $T - T_a/T_0 - T_a = 0.25, 0.50, 0.75, 0.99$. **b**, **d** Same as **a**, **c**, but in a time-averaged domain. Black and red lines represent the time-averaged velocity (vertical component) and temperature profiles, respectively. Profiles along the vertical axis are dashed, whereas horizontally integrated quantities are solid. All the values are normalized by using the initial conditions. The averaging window is from 500 to 1000 s. Symbols represent the uneroded ($H_{un}$; green square), collapse ($H_c$; red star), and jet ($H_{jet}$; blue circle) heights

## Table 1 Input parameters used for the end-members

| Variable | Partial collapse | Total collapse |
|---|---|---|
| Mass eruption rate (kg s$^{-1}$) | $10^8$ | $10^8$ |
| Temperature of the mixture (K) | 1123 | 1123 |
| Gas content (water content; wt%) | 2.0 | 1.0 |
| Vent pressure (MPa) | 10 | 0.1 |
| Vent mixture density (kg m$^{-3}$) | 964 | 19.3 |
| Vent mixture velocity (m s$^{-1}$) | 102 | 72 |
| Vent radius (m) | 18 | 151 |
| Inlet pressure (MPa) | 0.1 | 0.1 |
| Inlet mixture density (kg m$^{-3}$) | 9.64 | 19.3 |
| Inlet mixture velocity (m s$^{-1}$) | 203 | 72 |
| Inlet radius (m) | 127 | 151 |

ambient air and diluting, and depending on the amount of mixing, it may either be re-entrained into the eruptive column or remain denser than the atmosphere forming a PDC of small runout (<7 km). At the same time, the outer mixed portion of the column becomes buoyant and forms a convective plume that in this case reaches ~38 km (Methods section).

Color contours in the left-half of Fig. 1b display the time-averaged mixture temperature field, normalized to its initial value. The jet is characterized by a hot inner core surrounded by a region of gradually decreasing temperatures. Plots in the right-half of Fig. 1b depict time-averaged velocity and temperature profiles, along the vertical $z$-axis (black and red dashed lines) and integrated across horizontal cross-sections (black and red solid lines; cf. ref. [23]). These profiles allowed us to differentiate the height at which the vertical momentum along the $z$-axis is exhausted (uneroded height—$H_{un}$) from the one at which the horizontal average of the vertical momentum of the whole eruptive column is exhausted (jet height—$H_{jet}$). We computed $H_{jet}$ by looking at the minimum of the horizontally integrated velocity profile, although different alternatives have been proposed, either based on the generalized Morton's length scale[23,38,39] or on the Bernoulli's equation[13]. They are however consistent with the adopted one, differing by less than 10% (see Methods section). In this case, $H_{un} = 3.1$ km and $H_{jet} = 2.2$ km. According to refs. [11,13], the notable difference between $H_{un}$ and $H_{jet}$ is due to the effect of dilution associated with turbulent mixing, eroding the momentum flux from the jet edges. In other words, the higher the spread between these two heights, the

greater the gradual outer mixing of the erupted material with external air, and the stronger the decrease in temperature of the mixture. Finally, $Q_c$ is 15% of the mass leaving the source, and $T_c$ is estimated at 790 K.

Numerical results for the near-total collapse end-member reveal an internal structure of the jet characterized by a stable, symmetrically collapsing fountain (Fig. 1c, Supplementary Fig. 2c, d). Here, $H_{jet}$ and $H_{un}$ are very close to each other (~300 and ~380 m, respectively; Fig. 1d, right half), indicating that the mixing effect is not vigorous enough to erode the eruptive jet. Consequently, the mixture remains at a high temperature and high density until it reaches ~300 m where it starts to spread radially, forming a large-scale eddy (Fig. 1c, Supplementary Fig. 2c, d). Notwithstanding the ingestion of atmospheric air promoted by the large-scale eddy, the continuous feeding of mass from the collapsing jet does not allow the mixture to dilute, leaving it denser than the atmosphere, with solid mass fraction and temperature comparable to the inlet conditions. This end-member scenario results in a total collapse ($H_c$ ~300 m) with the generation of long runout PDCs (>20 km), whereas a small amount of material forms a buoyant vertical column that reaches a maximum height of ~19 km above the inlet. $Q_c$ is estimated at 79% of the mass leaving the inlet, and $T_c$ is 1050 K (Methods section). Given the low $H_c$, the jet does not have enough time to develop efficient turbulence and air entrainment, and this explains the high temperature of the collapsing mixture. Since turbulent fluctuations are closely related to dilution intensity, or equivalently to the entrainment rate, the dilution efficiency in the total collapse regime is weaker than the partial collapse end-member. Therefore, density and temperature fluctuations within the jet region are weaker in the case of total collapse. Indeed, the distribution of the time-averaged temperature (Fig. 1d) and density (Supplementary Fig. 2d) fields is similar to the instantaneous one (Fig. 1c, Supplementary Fig. 2c).

**Thermodynamic constraints for collapsing mixtures.** To better understand the mechanism of temperature drop due to mixing with atmospheric air, we here derive, from thermodynamic considerations, the thermal conditions that would characterize the collapsing mixture. We assume an isobaric transformation between an initial state where the erupted material and air are perfectly separated and a final state where the two phases are perfectly mixed and the resulting mixture is homogeneous. The final mixture temperature $T(\xi)$ can be expressed, from enthalpy conservation, as a function of the mass fraction of the erupted material $\xi$[13]. A dimensionless temperature $\theta(\xi)$ can be written as

$$\theta(\xi) \overset{\text{def}}{=} \frac{T(\xi) - T_{air}}{T_0 - T_{air}} = \frac{\xi C_{Pm}}{\xi C_{Pm} + (1-\xi)C_{Pair}}, \quad (1)$$

where $T_0$ is the initial temperature of the erupted material, $T_{air}$ is the initial air temperature, and $C_{Pair}$ is the specific heat at constant pressure of the air (thermodynamic properties used are listed in Supplementary Table 2). The specific heat at constant pressure of the erupted material is $C_{Pm} = y_w C_{Pw} + (1-y_w)C_s$, where $y_w$ is the initial gas fraction in the erupted material, $C_{Pw}$ and $C_s$ are the specific heats of the gas and solids, respectively. The ratio between the mixture density $\rho(\xi)$ and the air density $\rho_{air}$ is

$$\frac{\rho(\xi)}{\rho_{air}} = \frac{R_{air}T_{air}}{[\xi y_w R_w + (1-\xi)R_{air}]T(\xi)}, \quad (2)$$

where $R_w$ and $R_{air}$ are the gas constant of the erupted gas and air, respectively. Eqs. (1) and (2) are derived under certain assumptions. The enthalpy is conserved, thus the transformation is at constant pressure and the effects of dissipation and gravity can be disregarded. The erupted material in the final state is considered

as a homogeneous fluid made of a perfect mixture of gas and particles. Particle separation is thus ignored. Air temperature and density are constant, i.e., atmospheric stratification is neglected. Finally, the volume fraction of solid is negligibly small. In the following, we compare theoretical results with the 3D simulation, to evaluate to which extent these assumptions are correct cell by cell and in the whole collapsing region. From Eqs. (1) and (2), the minimum temperature and mass fraction for which the mixture is nonbuoyant ($\rho(\xi) > \rho_{air}$) can be derived

$$\theta_{min} = \frac{\alpha\beta\gamma - 1}{(\alpha - 1)\beta\gamma + \gamma}, \quad (3)$$

$$\xi_{min} = \frac{\beta\gamma - \alpha}{\gamma + 1 - \alpha} \quad (4)$$

$$\alpha = \frac{C_{Pm}}{C_{Pair}}; \ \beta = \frac{R_{air}}{R_{air} - y_w R_w}; \ \gamma = \frac{T_0}{T_{air}} - 1.$$

In our case, $\alpha \approx \beta \approx 1$ and Eqs. (1)–(4) simplify into

$$\theta(\xi) = \frac{\alpha\xi}{(\alpha - 1)\xi + 1} \approx \xi.$$

$$\frac{\rho(\xi)}{\rho_{air}} = \frac{\beta}{(\beta - \xi)(\gamma\theta + 1)} \approx \frac{1}{(1 - \xi)(\gamma\xi + 1)}.$$

$$\theta_{min} \approx \xi_{min} \approx \frac{\gamma - 1}{\gamma} = \frac{T_0 - 2T_{air}}{T_0 - T_{air}}.$$

As pointed out by ref. [13], $\theta_{min}$ and $\xi_{min}$ primarily depend on $T_0$ and $T_{air}$. Using $T_0 = 1100$ K and $T_{air} = 290$ K, we get $\theta_{min} \approx \xi_{min} \approx 0.64$. Below this value, a homogeneous mixture that cooled down by the sole effect of air entrainment would necessarily be positively buoyant.

In the volcanic case, stratification would be nonnegligible: the air density drop along the collapsing region can be as high as 40% and the temperature drop is about 10% (see Supplementary Fig. 1). In Fig. 2, we show the comparison between the theoretical relationships (1) and (2) (solid curves), and the simulated values of nondimensional temperature and density in each cell of the collapse region (points). The simulated dimensionless density is evaluated with Eq. (2), considering atmospheric temperature stratification $T_{air}(z)$ for each cell in the collapsing region, while its theoretical value is obtained using the average air temperature between the crater and the collapsing height. The two end-member cases, with and without particle decoupling, are compared. In the equilibrium case (dusty-gas; Fig. 2a, c), the thermodynamics of the numerical solution almost perfectly follows the simplified theoretical model in every cell of the collapsing region. This result is twofold: on one hand, it shows that all the hypotheses used to obtain Eqs. (1) and (2) are locally valid; on the other hand, it confirms the ability of the numerical solver to accurately solve the enthalpy equation. In the case of particle decoupling (Fig. 2b, d), both temperature and density values deviate from Eqs. (1) and (2), although following the expected theoretical trend. Since in this case we cannot use Eqs. (1) and (2) to define $\theta_{min}$ and $\xi_{min}$, these parameters are evaluated by looking for the last point with negative buoyancy (Fig. 2b, d). Particle separation makes $\theta_{min}$ and $\xi_{min}$ smaller than those predicted by Eqs. (3) and (4). Consequently, particle clustering and settling enhance $Q_c$ and the temperature drop. For the two end-members, we find $Q_c = 15\%$ and $Q_c = 79\%$ with particle decoupling, while $Q_c = 4\%$ and $Q_c = 70\%$ without. We quantify this effect in Supplementary Fig. 3, where a larger temperature drop is observed because of particle decoupling (up to 20% for the partial collapse end-member).

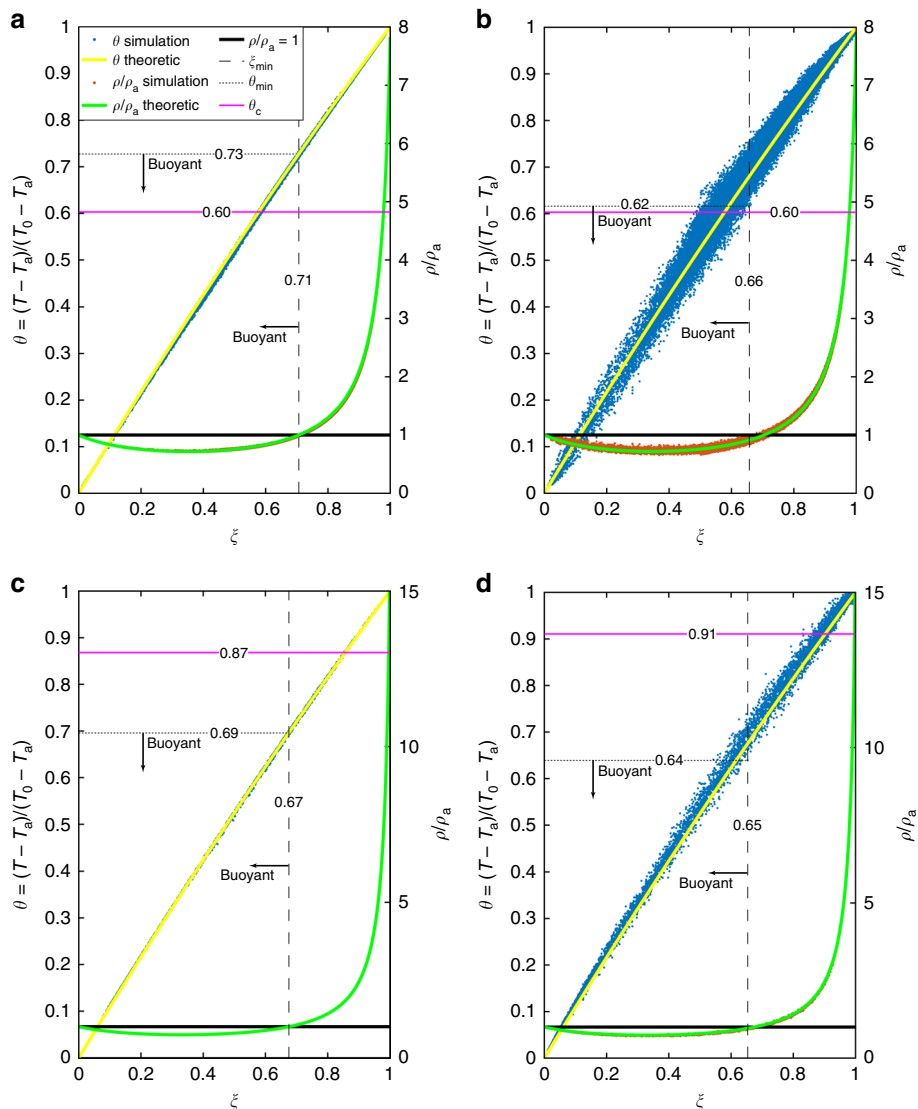

**Fig. 2** Thermodynamic properties of the collapse region for the reference models. Dimensionless temperature $\theta$ and density $\rho/\rho_a$ as a function of erupted material mass fraction $\xi$ in the collapsing region. Blue and red points are extracted from the numerical simulation data (temperature, mass fractions, and density), and the atmospheric temperature in each cell of the collapse region (see Supplementary Fig. 1). Yellow and green solid lines are obtained from Eqs. (1) and (2), using an average atmospheric temperature between the crater and the collapse height. The magenta solid line highlights the nondimensional value of the average temperature of the collapsing region $\theta_c$ (computed as described in the Methods section), while the black solid line is the level of neutral buoyancy. Black dotted and dashed lines indicate $\theta_{min}$ and $\xi_{min}$, respectively. All the blue points below $\theta_{min}$ and all the red points left of $\xi_{min}$ are less dense than the atmosphere at the same height. Panels **a** and **b** are for the partial collapse end-member, while **c** and **d** are for the total collapse. Panels **a** and **c** are for the dusty-gas model, while **b** and **d** take into account particle decoupling

The distribution of the points along all the theoretical curves of Fig. 2 clearly indicates that the collapsing region is highly heterogeneous, both in terms of particle concentration and temperature. Thus, even if Eqs. (1) and (2) hold locally, they cannot be used to estimate the global temperature and buoyancy of the collapsing region. To estimate the average temperature $T_c$ in the heterogeneous collapse region, we used instead a weighted average (described in the Methods section), based on the mass distribution in every cell. Its non-dimensional value is reported in Fig. 2 (red line) as $\theta_c$. In the partial collapse case (Fig. 2a, b), $\theta_c$ is smaller than $\theta_{min}$, due to the heterogeneity and enhanced entrainment in the collapsing stream. In the fountaining regime (Fig. 2c, d) $\theta_c > \theta_{min}$, clearly indicating the establishment of full collapsing conditions with a smaller entrainment efficiency.

Despite such complexities in the collapsing region, $T_c$ (or $\theta_c$) appears to be a good estimator for the average basal temperatures

of PDCs in the two end-member simulations. As shown in Supplementary Fig. 4, for the partial collapse case, $T_c = 790$ K and the average PDC temperature $T_{PDC} = 809$ K, while for the total collapse end-member $T_c = 1090$ K and $T_{PDC} = 1062$ K. The good agreement between these estimates confirms the validity of our approach.

## Discussion

The behaviors described above for the two end-members have been observed in all simulated plume scenarios. Figure 3a shows the variations of $H_{un}$, $H_{jet}$, and $H_c$ as a function of the calculated $Q_c$, for the whole set of numerical simulations with kinematic decoupling (Supplementary Table 1). The displayed variations indicate that the dynamics and efficiency of air entrainment drive the collapse regime[10,11,40]. When the erosion of the jet structure

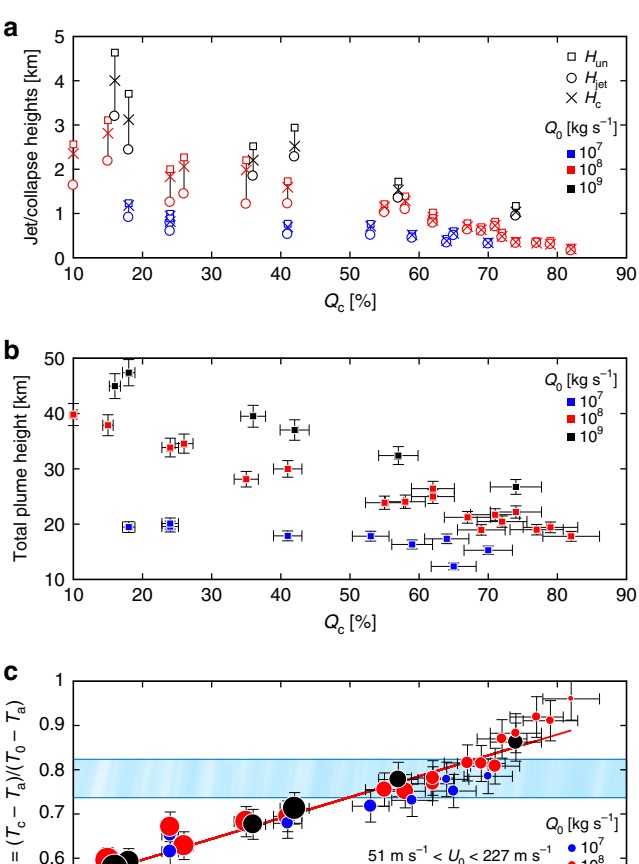

**Fig. 3** Summary of the numerical results. **a** Characteristic heights of the eruption jet as a function of the percentage of collapse $Q_c$ for all the simulated dynamics. Square symbols show the height at which the mean upward momentum of the central uneroded axis is exhausted ($H_{un}$). Circle symbols indicate the height at which the mean upward momentum of the whole eruptive column is exhausted ($H_{jet}$). Cross symbols represent the critical height at which the eruptive column starts to collapse ($H_c$; see Methods section). **b** Variation of the maximum plume height as a function of $Q_c$. **c** Relationship between $Q_c$ and its corresponding average temperature value $T_c$ normalized to the initial magmatic temperature $T_0$ with respect to the atmospheric air temperature $T_a$ ($\theta_c$). Blue, red, and black symbols in **a**–**c** represent the adopted mass eruption rate $Q_0$ at the source, and their size in **c** represents the relative exit velocity $U_0$. The upper and lower bounds of the light blue horizontal bar in **c** result from the ratio between the glass transition temperatures $T_g$ for rhyolitic and trachytic magmas, and their common eruptive temperature ranges. $T_g$ is taken anhydrous consistent with the initial condition of numerical experiments that involve the total volatile exsolution at vent. The red solid line shows the linear least square regression ($r^2 = 0.92$; $p$ value = $1.50 \times 10^{-18}$). A 10% of error is applied in panels **b** and **c** due to the mesh geometry (panel **a** is excluded for figure readability), as suggested by ref. [23]. Full references to the data sets used in the graph are reported in the Supplementary Table 1

until its upward velocity falls to zero (so that, $H_{un} \gg H_{jet}$), leading to a higher $H_c$ (<5 km). It follows that PDCs generated by low $H_c$ are fed by sustained high mass fluxes and can extend over distances much larger than those generated by high $H_c$. A corollary of our results is that progressively higher column collapse heights do not generate faster PDCs with longer runouts, in contrast with some previous views[41–43]. Our data indicate, instead, that long runouts are a direct consequence of high $Q_c$ at low $H_c$ (these results are consistent with refs. [44–46]).

The total plume height is generally related to the mass eruption rate using semi-empirical relationships based on buoyant plume theory[32,38,47]. Our findings indicate that an increase in $Q_c$ produces a substantial reduction of the total plume height for a given mass eruption rate (Fig. 3b), as the mass flux feeding the buoyant region is limited. For a mass eruption rate of $10^8$ kg s$^{-1}$, the total collapse of the column can lead to a maximum plume height of up to ~45% lower than that observed in the case of partial collapse, a value that increases even further when compared with fully buoyant plumes. Similar tendencies have been pointed out by previous numerical studies[14,16], which suggest that the variation in plume heights for a fixed mass eruption rate can be associated with different regimes of air entrainment due to fountain collapse. This evidence has important implications for the reconstruction of the mass eruption rate for those events in which a convective plume and a PDC phase coexist. Consequently, the maximum height reached by the column, and in turn, the inferred mass eruption rate, is not only affected by wind effects[48–50], but also by the collapse regimes.

We also observe (Fig. 3c) a linear correlation between $Q_c$ and its corresponding temperature drop. Thus, as the column regime changes from total to partial collapse, the average temperature of the mass that feeds a PDC decreases up to 45% of its initial eruptive value. This temperature drop is, for a given mass eruption rate, also related to an increase in the pyroclastic mixture upward velocity at the conduit exit. The correlation shown in Fig. 3c is relatively insensitive to the mass eruption rate adopted, implying that the thermal state of PDCs at their generation is dictated by $Q_c$, and hence depends on the amount of mixing with air during the column collapse process.

We illustrate the initial conditions that allow the collapsing mass to generate PDC deposits (also referred to as ignimbrites) hot enough to weld, by comparing the glass transition temperatures of erupted pyroclasts for a wide range of natural melt compositions[51] with their average eruptive temperatures (blue-shaded region in Fig. 3c). The resulting values indicate that only mixtures with a temperature decrease of up to ~26% of their eruptive temperature will thus meet the requirement for welding. These initial conditions provide a general framework for understanding the great variability of emplacement temperatures reported in literature for ignimbrites driven by the mechanisms of magmatic fragmentation (Fig. 4). Large volume ignimbrites (>1000 km³), which are those associated with supereruptions, are invariably largely welded[44], for which our quantitative study suggests initial conditions characterized by high $Q_c$ (>50%), low $H_c$ (<2 km) and consequently high temperatures (i.e., temperature drop <26% of the eruptive temperature). At decreasing volumes, ignimbrites can be entirely nonwelded with deposit temperatures that tend to be more scattered, although they exhibit a general trend of decreasing temperatures with decreasing volumes (Fig. 4). Our findings suggest that the relatively low temperatures of nonwelded ignimbrites (i.e., temperature drop >45% of the eruptive temperature) generated by magmatic fragmentation cannot be explained only by entrainment and mixing of atmospheric air during column collapse. The additional entrainment of ambient air into the jet region to levels sufficient to justify such low mixture temperatures will reflect the shift from a collapsing

by air entrainment is limited, ascending eruptive columns maintain their lateral flow characteristics, as revealed by $H_{un} \sim H_{jet}$, leading to a low $H_c$ (<2 km) and high $Q_c$ (>50%). Lower $Q_c$, on the contrary, are related to a more efficient mixing between the ejected mixture and the ambient air. In this case, the entrainment rapidly decelerates the upward velocity of the material within the annular mixing region, whereas the central unmixed region of the column continues to rise upward denser than the atmosphere

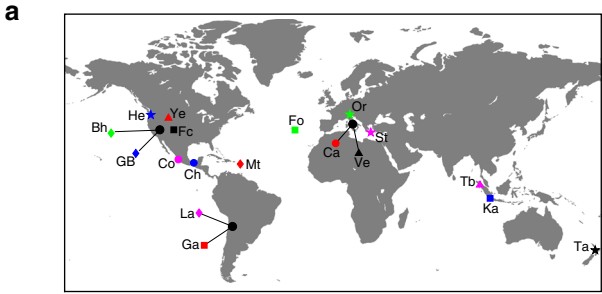

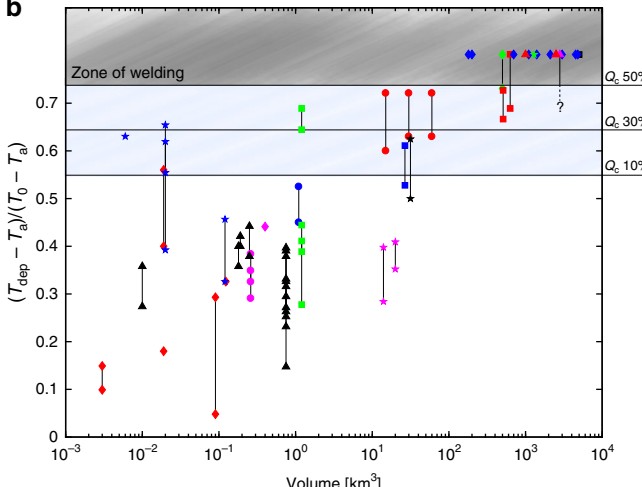

**Fig. 4** Collection of emplacement temperature data from historical column collapse-generated PDC deposits as a function of their respective estimated volume. **a** Volcanoes from which literature data were selected around the world. Abbreviations of volcano or ignimbrite names are: Bh = Bishop Tuff; Ca = Colli Albani; Ch = El Chichón; Co = Colima; Fc = Fish Canyon Tuff; Fo = Fogo; Ga = Cerro Galán; GB = Great Basin Ignimbrite province; He = Mt. St. Helens; Ka = Krakatau; La = Lascar; Mt = Montserrat; Or = Ora ignimbrite; St = Santorini; Ta = Taupo; Tb = Toba; Ve = Vesuvius; Ye = Yellowstone. Symbols correspond to each eruptive center. **b** The emplacement temperatures $T_{dep}$ are normalized by the average eruptive temperatures $T_0$, which are determined either from experimental data for the associated historical eruption or by using the corresponding typical eruption temperature based on melt composition, with respect to the atmospheric air temperature (assumed $T_a \approx 273\ K$). The percentage of collapse $Q_c$ is also illustrated. Ignimbrites deposited above the glass transition temperature $T_g$ are welded. Full references to the data sets used in the graph are reported in the Supplementary Table 3. Dome-generated and phreatomagmatic PDC deposits were not included in the analysis

to a fully buoyant column. Thus, at least one additional mechanism must have contributed to cool down the temperature of PDC deposits while, at the same time, maintaining the average buoyancy of the mixture negative. Clearly the average temperature of the collapsing mass depends on the initial eruptive temperature that varies also with the involvement, even weak, of groundwater or surface water (such as lakes, or glaciers), contributing to control the thermal evolution of the erupted mass[52]. On the other hand, air entrainment during partial collapse, with $Q_c$ ranging from 50 to 10%, is likely to be the primary mechanism that can justify nonwelded ignimbrites with intermediate emplacement temperatures. However, for these cases, we cannot quantify the possible contribution of mixing during lateral flow propagation on the final deposit temperature, as here we are not addressing heat loss during transport. A growing number of case studies show a decrease in the emplacement temperature along

flow of only a few tens of degrees[17,20,53], suggesting that ignimbrites can be thermally stable along their extent. Such temperature drop along flow is substantially smaller than what we observe in the collapse region. This implies that the primary controls on the emplacement temperature of nonwelded ignimbrites, at least for temperature drops between 26 and 45% of the eruptive temperature, are the decrease of $Q_c$ (<50%) and the increase of $H_c$ (>2 km).

In conclusion, our analysis of thermal collapsing regimes provides a new insight into the mechanism of mixing and partial collapse in turbulent volcanic columns, and gives a promising perspective on the potential of PDC emplacement temperatures to serve as markers for determining the column collapse conditions of both ancient and recent eruptions. This would be invaluable for improving models used for probabilistic hazard assessments[54,55] and the forecast of volcanic ash dispersal[48,49,56], as the initial physical properties of PDCs and the total plume height depend on collapse regimes.

## Methods

**Numerical model.** We used the numerical code ASHEE[23,24] to simulate the 3D dynamics of volcanic plumes. ASHEE solves the Eulerian compressible balance equations of mass, momentum and enthalpy of gas–particle mixtures, adopting the dynamic Large Eddy Simulation approach (see ref. [24] for the model equations). Gas and particles are treated as interpenetrating continua, with kinetic decoupling but thermal equilibrium among the gas and particulate phases. The fluid transport equations are discretized with a Finite Volume method, implemented in the open-source OpenFOAM framework for CFD (www.openfoam.org), by adopting a second-order accurate in time and space scheme to resolve turbulent fluctuations without excessive numerical diffusion. The computational grid is non-uniform and non-orthogonal, with the topological structure described by ref. [24] and has a minimum cell size of $r_0/8$ near the inlet region. The grid size increases with a radial and vertical grading factor of 1.0446 and 1.00187, respectively. The resulting total number of cells ranges from 714,240 to 2,642,944. The numerical solver is based on a semi-implicit segregated solution strategy with two PISO and two PIMPLE iterative loops[24].

**Derivation and constraints to input conditions.** We set the initial source conditions considering three scenarios, based on three different mass eruption rates ($10^7$, $10^8$, and $10^9\ kg\ s^{-1}$, Supplementary Table 1) at the conduit exit (vent). These scenarios make it possible to study the different regimes possibly occurring during explosive eruptions (also considering that eruptions at lower mass eruption rates usually generate no or minor pyroclastic density currents). All simulations are carried out with the constant mass eruption rate throughout the whole simulation time, within the same stratified atmosphere (Supplementary Fig. 1). To better understand the first-order effect of the initial source parameters, we assume, like most prior workers, windless atmospheric conditions only, although a strong wind field may in principle enhance the entrainment rate at the boundary of the eruptive jet, thus affecting the collapse regime.

For every given mass eruption rate, a set of independent mixture variables have been chosen to characterize the eruptive scenario by imposing the main flow parameters at conduit exit (vent), namely the magmatic temperature ($T$), the total water content ($y_w$) and the gas pressure ($P$). We preferred to impose the conditions at the conduit exit, instead of those at the atmospheric pressure inlet, because we believe that it is easier to constrain them from petrologic, experimental and numerical studies[11,57–59]. They are reported as independent variables in Supplementary Table 1. In addition, as often done in volcanic flow models, we have assumed sonic (choked-flow) conditions at the conduit top[13,60,61] as a constrain to vent velocity. To simplify the analysis, we have assumed that the mass fraction of solids is much larger than that of gases (which is valid in the investigated regimes, e.g., ref. [35]), so that we can write

$$U_v = c_{sound} \approx \sqrt{R_m T} \qquad (5)$$

where $U_v$ is the mixture velocity at the conduit exit, and the mixture gas constant is

$$R_m = y_w R_w$$

where $R_w$ is the gas constant of water vapor (thermodynamic properties used are listed in Supplementary Table 2), assumed to be the only magmatic gas in the eruptive mixture.

Above the vent, decompression of the mixture involves the formation of complex jet structures, which are largely influenced by the thermodynamic properties and by the geometry of the crater[14,31,34,36], and require a fine grid to be resolved numerically[35]. To overcome this difficulty, in this work we assume that the multiphase mixture has already decompressed in a crater above the vent. We compute analytically the flow conditions after decompression (dependent variables

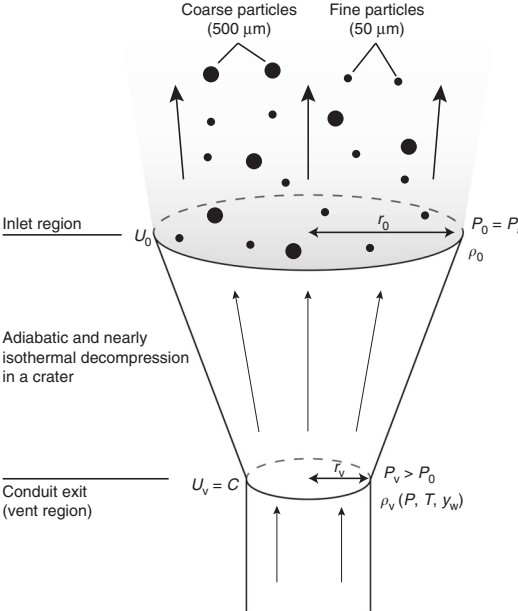

**Fig. 5** Numerical model setup. Schematic drawing of a volcanic jet, illustrating the definitions of the parameters before (vent region) and after decompression (inlet region) of the mixture, where $U_0$, $\rho_0$, and $r_0$ are the dependent source conditions of the numerical domain. The adiabatic decompression (from $P_v > P_0$) of the mixture is computed analytically assuming a correct expansion[13], in which all quantities are uniform (i.e., top-hat distribution) at the level of atmospheric pressure ($P_0 = P_a$). The flow temperature $T$ remains constant, whereas density $\rho_0$ decreases accordingly to the equation of state. The final velocity $U_0$ is computed by Eq. (6), and the final radius $r_0$ is imposed by the mass conservation

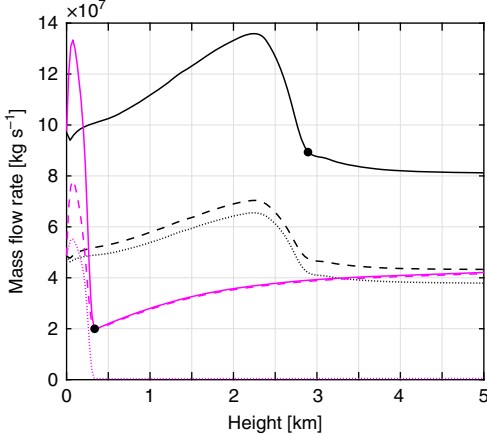

**Fig. 6** Mass flow rate patterns in case studies. Mass flow rates in the ascending column integrated over time and space (across horizontal cross-sections at different heights). Magenta and black lines indicate the mass flow rate of the near total collapse and partial column collapse end-members, respectively. Solid lines represent the sum of the mass flow rate of the coarse and fine particles. Dashed and dotted lines represent the mass flow rate of the fine and coarse particles, respectively. Black dots represent the height of collapse $H_c$

in Supplementary Table 1) by assuming an adiabatic and nearly isothermal transformation (i.e., the heat capacity ratio is ≈1; Fig. 5).

In such an overpressured flow, the flow accelerates due to the rapid adiabatic expansion. In this way, an average flow velocity $U_0$ is calculated with the model of ref. [33]

$$U_0 = U_v + \frac{P_v - P_0}{\rho_v U_v}, \tag{6}$$

where $P_0$ and $P_v$ are the atmospheric and mixture pressures at the vent, respectively, and $\rho_v$ is the mixture density before the decompression defined by the approximate equation of state (which is a good approximation for $y_w \gg 0.1\%$) as

$$\rho_v = \left( \frac{(1 - y_w)}{\rho_s} + y_w \frac{R_w T}{P_v} \right)^{-1} \approx \frac{P_v}{y_w R_w T}, \tag{7}$$

where $\rho_s$ is the density of the liquid magma.

The decompression, also, leads to the decrease of the mixture density, which is calculated assuming again an isothermal mixture

$$\rho_0 = \frac{\rho_v P_0}{P_v},$$

and an increase of the jet radius, which can be calculated as a function of the given mass eruption rate $Q$, velocity $U_0$ and density $\rho_0$ of the decompressed mixture

$$r_0 = \sqrt{\frac{Q}{\pi \rho_0 U_0}}.$$

Since velocity variations for pressure higher than about 10 MPa (i.e., a typical yield strength of rocks at shallow depth) are almost constant (velocity varies indeed approximately as $U_0/U_v = 2 - P_0/P_v$, accordingly to ref. [13]), we adopt three different mixture pressure at the vent (0.1, 0.2, and 10 MPa) to obtain a broad variability of mixture velocity. We also assume the case of a pressure-balanced sonic flow at the conduit exit. This might represent flow conditions occurring after some time from the beginning of the eruption, in a situation of open conduit system.

**Calculation of collapse and jet properties.** We provide here a description of the methodology used to quantify the properties of the collapsing column, namely the

percentage and the temperature of the collapsing mass. We also specify our experimental approach to identify the jet and plume heights. To smooth out fluctuations, all quantities are evaluated using time-averaged fields, in the time window between 500 and 1000 s.

An important parameter in forced plumes is the height where the flow moves from momentum dominated to buoyancy dominated[62]. Various definitions of this jet-plume transition height $H_{jet}$ can be used: one is the Morton length scale generalized to the non-Boussinesq multiphase case ($L_M$), which takes into account the effect of air entrainment in plumes[23,38,39,62]; another one is the definition ($H_{mmt}$) given by ref. [13], which disregards the effect of the entrainment. We decided to not use any of these definitions nor any assumption on entrainment and to use instead the minimum of the horizontally integrated velocity profile extracted from our numerical simulation results (blue circles in Fig. 1). It is similar to both previous definitions: for the partial collapse end member, $H_{jet} = 2.2$ km, $H_{mmt}$ defined by ref. [13] is 2.4 km, and $L_M = 2.1$ km when the jet entrainment coefficient is assumed to be 0.05[23,39]. The total (or maximum) plume height is defined as the highest atmospheric level of an assigned concentration threshold (1% of the tracer mass fraction at the inlet).

The procedure adopted to quantify the amount of collapsing mass is based on the computation of the mass flow rate across horizontal cross-sections of the ascending column, at different heights. The procedure is robust and holds for all regimes. The ascending column is defined as the part of the domain occupied by erupted material with vertical upward velocity $U_z > 0$. The edge between the inside and the outside of the column region is detected using the mass fraction of a tracer: the threshold is set to 1% of the tracer mass fraction at the inlet.

Figure 6 shows an example of the ash mass flow rate in the ascending column for simulations discussed in Fig. 1. An increase of the mass flow rate curve means that some ash is entering the ascending column. A decrease means that some ash is leaving the ascending column, feeding the collapse region. Analysis of the results reveals that the first peak is caused by the lower eddy formed in the collapse region. This recirculation eddy re-entrains collapsing mass into the ascending column, producing the observed increase of the mass flow rate. For both the two end-members, this effect involves approximately 30% of the mass eruption rate. Above this peak, the mass leaving the ascending column goes partly into the recirculation eddy and is partly lost by the collapsing region. The mass emerging above the collapsing region forms the buoyant region of the plume. The fraction of mass lost during this process is the collapsing fraction. It can be quantified by measuring the mass flow rate emerging above the collapse eddy. We, therefore, estimate the percentage of collapse $Q_c$ as the ratio between the total mass flow rate above the collapsing eddy and the mass flow rate imposed at the vent (Supplementary Table 1). The height at which this value is peaked represents the height of collapse ($H_c$; see black points in Fig. 6).

As we are interested in determining the mean temperature of the averaged mass that collapse from the eruptive column, we have tested different approaches to define the collapse region. Similarly to the definition given for the ascending column, the collapse region is the portion of the domain where the vertical component of the velocity is negative ($U_z < 0$), the tracer mass fraction ranges from

0.01 to 1 with respect to the inlet value and lays within a distance of 10 inlet radii ($r = 10r_0$). In this region, we calculate the mean temperature of the collapsing mass $T_c$ (Supplementary Table 1), as the weight average of the temperature with respect to the mixture density

$$T_c = \int_V \rho T \, dV / \int_V \rho \, dV.$$

## Data availability

Data that support the findings of this study are available in the paper or in the Supplementary Information. Further data are available on request from the corresponding author M.T.

## Code availability

ASHEE code is available from the author M.C. upon reasonable request.

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

## Acknowledgements
We thank Antonio Costa, Brittany Brand, and Pablo Tierz for discussions. In this study, we used the perceptually uniform color maps provided by Fabio Crameri. The computations of ASHEE were carried out on the LAKI HPC system at INGV Pisa. M.C. acknowledges CINECA award N. HP10BRDK2T (2017) for high-performance computing resources used for testing the ASHEE code. The INGV team has been partially supported by the Italian Ministry of University and Research (MIUR) through the FISR 2016 project - Centro di studio e monitoraggio dei rischi naturali dell'Italia centrale. The Grant to Department of Science, Roma Tre University (MIUR-Italy Dipartimenti di Eccellenza, Art. 1, com. 314–337 Legge 232/2016) is gratefully acknowledged.

## Author contributions
M.T. performed the numerical simulations and analyzed the data, conducted literature research, and wrote the first draft of the paper. M.C. developed the ASHEE code, prepared the simulations, and supported the definition and the analysis of the results. T.E.O. supported the definition of the numerical experiments and their implications. G.G. contributed to the volcanological implications. G.G. and T.E.O. conceived the project. All authors contributed in the design of the study, discussed the results, and commented on the paper.

## Additional information

**Competing interests:** The authors declare no competing interests.

