## [Peer Review File · Nature Communications]

Reviewers' comments:

Reviewer #1 (Remarks to the Author):

Review of 'Thermal regime of eruption column collapses and generation of hot pyroclastic flows'

Erika Rader – University of Idaho

This study uses a 3-d numerical model to simulate heights and convection processes within volcanic eruption columns with the intention of constraining the conditions that lead to column collapse and PDC formation. They found that reduced entrainment of ambient air leads to a hot, unstable column which can travel very far when it collapses. This variable is more likely to produce a long run-out from a PDC than the mass flux, which was previously thought to be the most significant factor for eruptions like Taupo. The conclusions are interesting and valuable to the volcanologic community and their work is convincing, although more discussion of what the modeling terms mean in real world scenarios (eg. What does a density of 19.3 kg/m³ mean and is it found in nature? See comment for Line 159 also) would make the results more impactful. There is not enough information in this manuscript for a non-expert to repeat the study, however, they list other studies which have more detailed methods sections, suggesting one could work it out if need be.

Lines 113- 118. It is unclear to me the exact comparison you are making. The second sentence says a 'small amount of material becomes buoyant and reaches 19 km' is this small amount of material reaching 19 km in a vertical plume? Or does it only run out 19 km compared to the majority of the material which runs out 20 km?

Line 119 – fluctuations of what?

Line 127 – might be good to state what the "whole set of numerical simulations" includes here.

Line 138 – seem to be missing some word here. Not clear what is contrasting with the idea, or what the idea is exactly.

Line 145 – perhaps this isn't something to be added to the manuscript, but how did you test how atmospheric conditions might affect the likelihood of air entrainment? What conditions support producing these hotter PDC run outs?

Line 157 – I don't believe the citation you provided for this line states that the glass transition is going to be 74-82% of the eruption temperature. You are assuming that eruption temperature is the same as the experimental temperature Giordano used in that paper. Perhaps clarify this point a bit so someone doesn't assume volcanos only ever erupt the temperature where glass of that composition is a crystal-free bubble-free liquid, as they were in the experiments.

Line 159 – Your model provides no data in Figure 3. That appears to be literature search with no analysis using the model. Why not include estimates of air entrainment for each, or column densities for each or some output which your model gives by matching the temperature and run-out length of the deposit?

Line 301 – Include 'Qc' in the caption.

Line 322 – add 'symbols correspond to each eruptive center' to caption, or something similar.

Review report on “Thermal regime of eruption column collapses and generation of hot pyroclastic flows” by Trolese et al.

This paper provides interesting and thoughtprovoking numerical results focusing on collapsing eruption columns, and discusses their applications to field data concerning temperatures of pyroclastic density currents (PDCs). The subject is important and some of the present results are novel. The present numerical results support the column collapse condition which has been recently proposed by Koyaguchi and Suzuki (2018) and Koyaguchi et al., (2018) (see Appendix of this review report). In addition, the results have quantitatively shown how the partial collapse develops in the transitional state of column collapse, which is certainly novel. I agree with the authors’ conclusion that the present study provides a new insight into the mechanism of partial collapse of eruption column. On the other hand, there seem to be several points to be improved in this paper; some of the analyses of the simulation results look still preliminary and do not fully explain the observations in nature (e.g., those in Figure 3). Accordingly, I recommend the publication of this paper after some revisions. The points that can be improved are listed below.

Major comment

Figure 2c shows that the proportion of the collapsed flow, Q_c , is correlated with the temperature of collapsing downflow, $(T_c - T_a)/(T_0 - T_a)$. This is an interesting observation, and it is the key idea of this paper to explain the variation in temperature of PDCs in nature (i.e., Figure 3). However, the physics behind the relationship in Figure 2c is not clearly presented in the paper. As a consequence, a part of the geological implication using Figure 3 is not convincing as it stands.

In fact, there is one important unresolved problem in Figure 2c. That is, *how can the mixture with low $(T_c - T_a)/(T_0 - T_a)$ be denser than air to form a PDC?* The variations in Q_c and $(T_c - T_a)/(T_0 - T_a)$ observed in Figure 2c reflect the variation in the mass fraction of ejected magma (i.e., mixing ratio of ejected magma and air) in the collapsed flow (let us define this mass fraction as ξ); Q_c and $(T_c - T_a)/(T_0 - T_a)$ decrease as ξ decreases. The problem is that the mixture with low ξ is inevitably buoyant and it cannot form a PDC.

Let us consider a homogeneous mixture with ξ as a reference state. For simplicity, it is assumed that the effect of particle separation is negligible. In this reference case, the temperature of the eruption cloud, T , is calculated from

$$\frac{T - T_a}{T_0 - T_a} = \frac{\xi C_{Pm}}{\xi C_{Pm} + (1 - \xi) C_{Pair}}, \quad (1)$$

where C_{Pair} and C_{Pm} are the specific heats of the air and the magma at constant pressures, respectively, and T_a and T_0 are the temperatures of the ambient air and the magma, respectively. The specific heat of the magma is calculated from those of pyroclasts (C_s) and volcanic gas (C_{Pg}) as $C_{Pm} = y_w C_{Pg} + (1 - y_w) C_s$. This equation means that the vertical axis of Figure 2c is closely correlated with ξ (see the black curve in Figure 1 in this review report). On the other hand, substituting T in Eq.(1) into the equation of state, we can calculate the density of the mixture ρ as

$$\rho \approx \frac{P}{\{\xi y_w R_w + (1 - \xi) R_{air}\} T}, \quad (2)$$

Figure 1: The normalized temperature (black curve) and the density (ρ ; red curve) as a function of the mass fraction of ejected materials in the eruption cloud (ξ) based on Eqs. (1) and (2). $T_0 = 1123\text{ K}$ and $y_w = 0.02$ are assumed. The horizontal thin red line indicates the density of air at $p = 1.013 \times 10^5\text{ Pa}$ (ρ_{air}).

where P is the pressure, R_{air} is the gas constant of the air. Here, we used the approximation that the volume fraction of pyroclasts is negligibly small in the eruption cloud. From Eq.(1) and Eq.(2) we obtain the relationship between the mass fraction of the ejected material ξ and the density ρ as the red curve in Figure 1. My preliminary calculation in Figure 1 suggests that the mixture with $(T_c - T_a)/(T_0 - T_a) < 0.7$ is expected to be buoyant (Figure 1).

This simple consideration based on the reference state leads to a conclusion to explain the occurrence of low temperature PDC in Figure 3. The values of T and ξ for $\rho = \rho_{\text{air}}$ (the star in Figure 1) primarily depend on T_0 ; they decrease as T_0 becomes low, as is the case of phreatomagmatic eruptions (see Koyaguchi and Suzuki (2018) for more comprehensive discussions). Thus, it is inferred that very low temperature PDCs in Figure 3 resulted from phreatomagmatic eruptions with low T_0 .

Figure 2c in this paper, on the other hand, indicates that the variation in PDCs' temperature can result from variation in U_0 (i.e., initial momentum) for given mass eruption rate Q_0 . As U_0 increases for given Q_0 , Q_c decreases (see Appendix below), which generates a PDC with a low $(T_c - T_a)/(T_0 - T_a)$. To apply this result to the observations in nature (e.g., the observations in Figure 3), we must specify the minimum value of $(T_c - T_a)/(T_0 - T_a)$ for the mixture to be denser than air to form a PDC (i.e., whether it is as low as 0.5 like the case of Figure 2c or whether it can be even lower). For this purpose, we must understand the mechanism how the mixtures with $(T_c - T_a)/(T_0 - T_a) < 0.7$ can be denser than air. There may be some possible mechanism to form a dense mixture with low $(T_c - T_a)/(T_0 - T_a)$ as follows.

- The mixture of the erupted material and air is not necessarily a consequence of

simple binary mixing, which is assumed in the calculations of Eqs. (1) and (2), but may be heterogeneous. If the thermal equilibrium in the mixture is incomplete, the density of such heterogeneous mixture may deviate from the red curve in Figure 1 to considerable extent (see the light blue region in Figure 1 in this report).

- The above calculations of Eqs. (1) and (2) is based on the assumption that the effect of the particle separation is negligible, whereas Supplementary Figure 3 indicates the effect of the separation of coarse pyroclasts may be significant.

To understand the implication of Figure 2c, we must somehow evaluate the effects of the above mechanisms. Particularly, to clarify the effect of the first mechanism, it is essentially important to describe the heterogeneity of the collapsing down-flow in the simulation results. For example, according to the definition in line 112 of Supplementary information, T_c is calculated as an averaged temperature; however, it is not clear from the averaged value whether the collapsing down-flow is composed of a uniform mixture with a uniform temperature of T_c or it is composed of heterogeneous mixture with a wide range of T . It is also helpful if we could know the variations in the temperature T , the density ρ and the mass fraction of ejected material ξ at each grid point before averaging; those values of T and ρ can be plotted as a function of ξ in a diagram like Figure 1 in this review report.

I suggest that the geological implication using Figure 3 is possible only after the above discussions on the mixture with low $(T_c - T_a)/(T_0 - T_a)$. Judging from Figure 2c, I have a feeling that the above two mechanisms may explain a dense mixture with $(T_c - T_a)/(T_0 - T_a)$ as low as 0.5; however, to explain the occurrence of PDCs with very low temperature (those of $T = 100\text{--}300^\circ\text{C}$ in Figure 3), additional effects such as the presence of ground water would be necessary.

Specific comments

Some other specific comments follow.

- (1) Lines 62–67: From the description of *Supplementary Methods* I could infer that the 3-D numerical simulations in this study are actually those for pressure-balanced jets with variable U_0 (i.e., so-called correctly expanded jets); however, the main text (as well as Table 1) gives an impression that the simulations are done for variable U_v and P_v , which is misleading. In this study, the physical quantities at $P = P_0$ (atmospheric pressure) are assumed to be uniform (i.e., top-hat distributions). On the other hand, recent numerical studies (Carcano et al., 2013; Koyaguchi et al., 2018) indicate that, when the jet is not pressure-balanced at the vent, the decompression and compression processes just above the vent strongly modify the distributions of the physical quantities at $P = P_0$. As is discussed in *Major comment*, the distributions of the physical quantities in the column (e.g., the heterogeneity of T) is considered to be critical to generate a dense mixture with a relatively low $(T_c - T_a)/(T_0 - T_a)$. To avoid readers' misunderstanding, the author should explicitly state in the main text that the 3-D numerical simulations in this study are essentially those for pressure-balanced jet with variable U_0 , and, if possible, discuss

how the assumption of top-hat distributions of physical quantities at $P = P_0$ affects the main conclusions of this paper.

- (2) Lines 95–97: The definition of partial collapse based on Q_c is rather *indirect* and it requires more explanations. I agree that Q_c represent something important for understanding of partial collapse, but I am not fully convinced that Q_c represents the fraction of collapsing mass as a PDC. There are at least two ambiguities. The separation of particles results in mass loss from an ascending column. Therefore, strictly speaking, the difference between the mass flow rate at the black dot and the total mass eruption rate does not represent the collapsing mass flow rate as a PDC. Considering the presence of large scale eddy around the top of fountain (H_c), a part of erupted mass may also be trapped by the large scale circulation even though the whole down-flow is not denser than the ambient air. These effects of the particle separations and circulations should be quantitatively evaluated to understand the meaning of Q_c . In my opinion, to clarify whether Q_c definitely represents the fraction of collapsing mass as a PDC, it is essentially important to show that the collapsing down flow is denser than the ambient air.
- (3) Line 103: What is the definition of H_{jet} ? Does it the level of the minimum point of horizontally integrated velocity in Figure 1? If so the level does not necessarily represent the level where the vertical momentum of the whole eruptive column is exhausted. The profile of horizontally integrated velocity is determined by competing effects of decreasing momentum due to negative buoyancy and increasing momentum of positive buoyancy. If the effect of increasing buoyancy due to positive buoyancy is large, the upward velocity may have a minimum point at a lower level before the initial momentum is exhausted.
- (4) The result in Figure 2b (and related text from line 141 to 145) is interesting, although it may not be the main subject of this paper. In fact, similar tendencies have been pointed out in Costa et al. (2018) and Koyaguchi et al. (2018). Although the present result in Figure 2b is more comprehensive and looks more systematic, those previous works can be cited here.
- (5) The caption of Supplementary Figure 3 needs to be corrected. There seem to be two important typos. Judging from the value of H_c (i.e., the positions of the black dots), the black curves are the results of partial collapse and the pink curves are the results of the near total collapse. In the pink curves, the dotted curve shows nearly zero mass flow rate at high altitudes. According to the caption, the upper part of the this column (with pink color) is composed of coarse particles shown by dashed curve alone and almost fully depleted in fine particles, which does not make sense to me. I suspect that dashed and dotted curves represent the mass flow rate of fine and coarse particles, respectively.

Other minor points are annotated in the pdf file.

Appendix: Consistency and discrepancy with the column collapse condition

proposed by Koyaguchi and Suzuki (2018) and Koyaguchi et al. (2018)

In the course of review process, I have confirmed how the present numerical results are consistent with the theoretical and numerical column collapse condition by Koyaguchi and Suzuki (2018) and Koyaguchi et al. (2018). Koyaguchi and Suzuki (2018) have shown that the column collapse condition of a pressure-balanced jet (e.g., Bursik and Woods, 1991) is expressed by a simple relationship as

$$M_a/\tilde{Q}^{1/5} = 1, \quad (3)$$

where M_a is the Mach number after decompression ($U_0/\sqrt{y_w R_w T_0}$ using the present notation) and \tilde{Q} is the dimensionless mass eruption rate (see Koyaguchi and Suzuki (2018) for the normalization factor). An eruption column is stable to form a buoyant plume when $M_a/\tilde{Q}^{1/5} > 1$, whereas it collapses to form PDCs as $M_a/\tilde{Q}^{1/5} < 1$.

Using the values in Table 1, we can plot Q_c (%) as a function of $M_a/\tilde{Q}^{1/5}$ for the present numerical results (Figure 2). The present results are consistent with the prediction by Koyaguchi and Suzuki (2018) in the sense that $Q_c = 0$ (i.e., fully buoyant) for $M_a/\tilde{Q}^{1/5} > 1$. The present results also show that Q_c gradually increases with decreasing $M_a/\tilde{Q}^{1/5}$ in the region of $M_a/\tilde{Q}^{1/5} < 1$. It is interesting to see the variation of Q_c with $M_a/\tilde{Q}^{1/5}$ collapses to a single relationship for $Q_0 = 10^7$ – 10^9 kg/s; this may suggest that the present numerical study has captured a novel universal feature of the transition from a buoyant plume to total collapse.

It should be noted that the present numerical results and those of Koyaguchi et al. (2018) do not perfectly coincide with each other in a quantitative sense. The transitional state (i.e., partial collapse etc) is observed in the range of $M_a/\tilde{Q}^{1/5} < 1$ in the present study. On the other hand, it is typically observed in the range $1 < M_a/\tilde{Q}^{1/5} < 1.2$ in the 3-D simulations of Koyaguchi et al. (2018); the range of the transitional zone varies with Q_0 and P_v to a considerable extent in their simulations. This quantitative discrepancy may be caused by the fact that the present simulations take the effects of particle settling into account, whereas those of Koyaguchi et al. (2018) do not. The discrepancy may partly result from the difference in the condition at the level of $P = P_0$. In Koyaguchi et al. (2018), the boundary condition of $P_v \neq P_0$ is set at the crater top such that the physical quantities at the level of $P = P_0$ is heterogeneous because of the decompression/compression processes; this heterogeneity causes the diverse features in the transitional state in their simulations. In this study, on the other hand, a top hat distribution is imposed at the level of $P = P_0$. It is suggested that further study would be necessary to fully understand the diverse features in the transitional state.

References

- [1] Bursik, M. I., and A. W. Woods (1991), Buoyant, superbuoyant and collapsing eruption columns, *Journal of Volcanology and Geothermal Research*, 45, 347–350, doi.org/10.1016/0377-0273(91)90069-C.

Figure 2: Q_c (%) as a function of $M_a/\tilde{Q}^{1/5}$. For the definition of M_a and $\tilde{Q}^{1/5}$ see text. The color of symbols are the same as the manuscript. Green: $Q_0 = 10^7$ kg/s, red: 10^8 kg/s, and black: 10^9 kg/s.

- [2] Carcano, S., L. Bonaventura, T. Esposti Ongaro, and A. Neri (2013), A semi-implicit, second-order-accurate numerical model for multiphase underexpanded volcanic jets, *Geoscientific Model Development*, 6, 1905–1924, doi10.5194/gmd-6-1905-2013.
- [3] Costa, A, Y. J. Suzuki, and T. Koyaguchi (2018), Understanding the dynamics of explosive super-eruptions, *Nature Communications*, 9, 654, doi:10.1038/s41467-018-02901-0.
- [4] Koyaguchi, T., and Y. J. Suzuki (2018), The condition of eruption column collapse: Part 1. A reference model based on analytical solutions, *Journal of Geophysical Research*, 123, doi.org/10.1029/2017JB015308.
- [5] Koyaguchi, T., Y. J. Suzuki, K. Takeda, and S. Inagawa (2018), The condition of eruption column collapse: Part 2. Three-dimensional (3D) numerical simulations of eruption column dynamics, *Journal of Geophysical Research*, 123, /doi.org/10.1029/2017JB015259.

We wish to thank the Reviewers for their constructive feedbacks that greatly helped to improve the manuscript. Below we provide our replies in blue (bold) to all the Reviewer comments

Reviewer #1 – Erika Rader

This study uses a 3-d numerical model to simulate heights and convection processes within volcanic eruption columns with the intention of constraining the conditions that lead to column collapse and PDC formation. They found that reduced entrainment of ambient air leads to a hot, unstable column which can travel very far when it collapses. This variable is more likely to produce a long run-out from a PDC than the mass flux, which was previously thought to be the most significant factor for eruptions like Taupo. The conclusions are interesting and valuable to the volcanologic community and their work is convincing, although more discussion of what the modeling terms mean in real world scenarios (eg. What does a density of 19.3 kg/m³ mean and is it found in nature? See comment for Line 159 also) would make the results more impactful. There is not enough information in this manuscript for a non-expert to repeat the study, however, they list other studies which have more detailed methods sections, suggesting one could work it out if need be.

We sincerely thank the reviewer, Dr Erika Rader, for her positive comments about our manuscript. As we now specify in the revised manuscript (lines 77-78/373-376, manuscript with highlighted changes), the input parameters we have applied to our simulations were selected to be as realistic as possible, based on the currently available data range for Plinian eruptions (Carazzo et al., 2008; Mastin et al., 2009; Koyaguchi et al., 2010). Our initial source parameters can indeed produce fully realistic eruption scenarios leading to column collapse.

Concerning the lack of information, we prefer to not provide an accurate description of the fluid dynamical model, as the numerical code has been already extensively described (and tested against numerical and experimental benchmarks) in previous papers, which we included as references in our manuscript. We rather choose to focus on describing the adopted initial and boundary conditions, and on defining the variables used to investigate the collapse dynamics, so as readers can easily assess our experimental procedure.

Carazzo, G., Kaminski, E. & Tait, S. On the dynamics of volcanic columns: A comparison of field data with a new model of negatively buoyant jets. J. Volcanol. Geotherm. Res. 178, 104–115 (2008).

Mastin, L. G. et al. A multidisciplinary effort to assign realistic source parameters to models of volcanic ash-cloud transport and dispersion during eruptions. J. Volcanol. Geotherm. Res. 186, 10–21 (2009).

Koyaguchi, T., Suzuki, Y. J. & Kozono, T. Effects of the crater on eruption column dynamics. J. Geophys. Res. Solid Earth 115, (2010).

Lines 113- 118. It is unclear to me the exact comparison you are making. The second sentence says a ‘small amount of material becomes buoyant and reaches 19 km’ is this small amount of material reaching 19 km in a vertical plume? Or does it only run out 19 km compared to the majority of the material which runs out 20 km?

We apologize for not being clear enough at this point. The intention was to say that a small amount of material rises buoyantly through the atmosphere as a vertical plume up to about 19 km, whereas 79% of the erupted material collapses to form a pyroclastic flow of long runout (>20 km). We have updated this information in the manuscript (lines 161-163, manuscript with highlighted changes).

Line 119 – fluctuations of what?

We simply meant density and temperature fluctuations, we have reworded accordingly (lines 169-174, manuscript with highlighted changes).

Line 127 – might be good to state what the “whole set of numerical simulations” includes here.

We are referring to all the simulations that are listed in Supplementary Table 1. We have added “(Supplementary Table 1)” to the sentence to specify this (lines 246-248, manuscript with highlighted changes).

Line 138 – seem to be missing some word here. Not clear what is contrasting with the idea, or what the idea is exactly.

We revised this sentence so it now reads, “A corollary of our results is that progressively higher column collapse heights do not generate faster PDCs with longer runouts, in contrast with some previous views³⁸⁻⁴⁰”. See lines 260-262, manuscript with highlighted changes.

Line 145 – perhaps this isn’t something to be added to the manuscript, but how did you test how atmospheric conditions might affect the likelihood of air entrainment? What conditions support producing these hotter PDC run outs?

We thank Erika Rader for her comment. We agree that different meteorological conditions may in part affect the efficiency of air entrainment during the jet phase, although it has recently been shown (e.g. Costa et al., 2016; Suzuki and Koyaguchi 2015) that, for example, atmospheric winds are mostly important in determining the total plume height for an assigned mass eruption rate, as they can shift downwind the buoyant and umbrella regions of the plume. This is particularly true in the case of weak plumes (mass eruption rate lower than $10^7 - 10^8$ kg/s), where the wind speed is comparable to the plume velocity. Given that our manuscript is the first study showing the thermal patterns leading to different column collapse regimes, we think that it is better and clearer to not consider the effects of wind, which will be investigated in future trials, as it is an interesting area for further extension. We decided to focus our study on the effect of the eruptions source parameters on the collapse efficiency, as we expect that those have a primary effect on the jet dynamics. For these reasons, our simulations are carried out in a stratified windless atmospheric condition (we have provided a new figure in Supplementary Materials to show the atmospheric profiles used for the simulations). Nonetheless, we expect the wind to have an effect too, even if weaker. Qualitatively, we expect an enhanced entrainment rate in

strong wind conditions. It is worth noting that we also consider ash kinematic decoupling and preferential concentration of particles (e.g. Cerminara et al., 2016) to take into account the effect of coarse ash on the collapsing dynamics, and hence on the mixing efficiency. In the revised version, we highlight all the previous points (please refer to lines 70-72/366-369 in the manuscript with highlighted changes).

Cerminara, M., Esposti Ongaro, T. & Neri, A. Large Eddy Simulation of gas–particle kinematic decoupling and turbulent entrainment in volcanic plumes. J. Volcanol. Geotherm. Res. 326, 143–171 (2016).

Costa, A. et al. Results of the eruptive column model inter-comparison study. J. Volcanol. Geotherm. Res. 326, 2–25 (2016).

Suzuki, Y. J. & Koyaguchi, T. Effects of wind on entrainment efficiency in volcanic plumes. J. Geophys. Res. Solid Earth 120, 6122–6140 (2015).

Line 157 – I don't believe the citation you provided for this line states that the glass transition is going to be 74-82% of the eruption temperature. You are assuming that eruption temperature is the same as the experimental temperature Giordano used in that paper. Perhaps clarify this point a bit so someone doesn't assume volcanos only ever erupt the temperature where glass of that composition is a crystal-free bubble-free liquid, as they were in the experiments.

We apologise for making this point not sufficiently clear. We did not use the liquidus temperatures of Giordano et al., 2005 as eruption temperature values, but only their glass transition temperatures. What we meant is that we use the ratio between the glass transition temperature (experimentally determined by Giordano et al. 2005) and the average magmatic/eruption temperature for rhyolitic and trachytic magmas (so that we consider a wide range of natural melt compositions and temperature) to find the values between which an erupted mixture is able to form a welded pyroclastic flow deposit. We have reworded the manuscript (please refer to lines 286/293 in the manuscript with highlighted changes) to make this point clearer.

Giordano, D., Nichols, A. R. L. & Dingwell, D. B. Glass transition temperatures of natural hydrous melts: a relationship with shear viscosity and implications for the welding process. J. Volcanol. Geotherm. Res. 142, 105–118 (2005)

Line 159 – Your model provides no data in Figure 3. That appears to be literature search with no analysis using the model. Why not include estimates of air entrainment for each, or column densities for each or some output which your model gives by matching the temperature and run-out length of the deposit?

Figure 3 (now Figure 4) shows a compilation of emplacement temperatures and volumes for several pyroclastic flow deposits reported in the literature, in order to appreciate the strong variability in temperatures and that such variability decreases at increasing erupted volumes, becoming progressively closer to welding temperatures.

We explain this observation as a function of column collapse regimes. Since the original figure could not be directly fed with data from our numerical experiments, as suggested by Reviewer

#1, we have edited this figure, by representing the temperature of the deposit proportionally to the eruptive temperature in order to include the percentage of collapsing mass derived by our models, which is linearly related to the temperature drop of the collapsing mass. We have therefore made changes to the explanatory text of the new version of Figure 4 at lines 293-328 in the manuscript with highlighted changes, reinforcing our findings, and explaining that: 1) the low mixing associated with near total collapse regimes fully justify welded pyroclastic flow deposits; 2) the highest mixing associated with partial collapse regimes cannot alone justify the occurrence of very low temperature pyroclastic flow deposits, and so it is necessary to consider at least an additional mechanism such as the involvement of external water during an eruption; 3) mixing associated with partial collapse - with percentages of collapse between 50% and 10% - can likely be the principal factor to justify unwelded pyroclastic flow deposits with intermediate emplacement temperatures.

Line 301 – Include ‘Qc’ in the caption.

This has been added in the caption.

Line 322 – add ‘symbols correspond to each eruptive center’ to caption, or something similar.

This has been added in the caption.

Reviewer #2

This paper provides interesting and thoughtprovoking numerical results focusing on collapsing eruption columns, and discusses their applications to field data concerning temperatures of pyroclastic density currents (PDCs). The subject is important and some of the present results are novel. The present numerical results support the column collapse condition which has been recently proposed by Koyaguchi and Suzuki (2018) and Koyaguchi et al., (2018) (see Appendix of this review report). In addition, the results have quantitatively shown how the partial collapse develops in the transitional state of column collapse, which is certainly novel. I agree with the authors’ conclusion that the present study provides a new insight into the mechanism of partial collapse of eruption column. On the other hand, there seem to be several points to be improved in this paper; some of the analyses of the simulation results look still preliminary and do not fully explain the observations in nature (e.g., those in Figure 3). Accordingly, I recommend the publication of this paper after some revisions. The points that can be improved are listed below.

We sincerely thank Reviewer #2 for the very helpful and encouraging comments. We appreciate his/her suggestions for improving the clarity and impact of our study. Overall, we recognize based on the Reviewer’s comments that we could have described more the thermodynamic properties of the collapse region, which are important to understand the collapse mechanisms and, in turn, the variation in PDC’s emplacement temperatures observed in nature. We have strived to address all the Reviewer’s concerns and summarize them below.

Major comment

Figure 2c shows that the proportion of the collapsed flow, Q_c , is correlated with the temperature of collapsing downflow, $(T_c - T_a) = (T_0 - T_a)$. This is an interesting observation, and it is the key idea of this paper to explain the variation in temperature of PDCs in nature (i.e., Figure 3). However, the physics behind the relationship in Figure 2c is not clearly presented in the paper. As a consequence, a part of the geological implication using Figure 3 is not convincing as it stands.

In fact, there is one important unresolved problem in Figure 2c. That is, *how can the mixture with low $(T_c - T_a) = (T_0 - T_a)$ be denser than air to form a PDC?* The variations in Q_c and $(T_c - T_a) = (T_0 - T_a)$ observed in Figure 2c reflect the variation in the mass fraction of ejected magma (i.e., mixing ratio of ejected magma and air) in the collapsed flow (let us define this mass fraction as ξ); Q_c and $(T_c - T_a) = (T_0 - T_a)$ decrease as ξ decreases. The problem is that the mixture with low ξ is inevitably buoyant and it cannot form a PDC.

Let us consider a homogeneous mixture with ξ as a reference state. For simplicity, it is assumed that the effect of particle separation is negligible. In this reference case, the temperature of the eruption cloud, T , is calculated from

$$\frac{T - T_a}{T_0 - T_a} = \frac{\varepsilon C_{Pm}}{\varepsilon C_{Pm} + (1 - \varepsilon) C_{P\text{air}}}; \quad (1)$$

where $C_{P\text{air}}$ and C_{Pm} are the specific heats of the air and the magma at constant pressures, respectively, and T_a and T_0 are the temperatures of the ambient air and the magma, respectively. The specific heat of the magma is calculated from those of pyroclasts (C_s) and volcanic gas (C_{Pg}) as $C_{Pm} = \gamma_w C_{Pg} + (1 - \gamma_w) C_s$. This equation means that the vertical axis of Figure 2c is closely correlated with ξ (see the black curve in Figure 1 in this review report). On the other hand, substituting T in Eq.(1) into the equation of state, we can calculate the density of the mixture ρ as

$$\rho \approx \frac{P}{\{\varepsilon \gamma_w R_w + (1 - \varepsilon) R_{\text{air}}\} T'} \quad (2)$$

where P is the pressure, R_{air} is the gas constant of the air. Here, we used the approximation that the volume fraction of pyroclasts is negligibly small in the eruption cloud. From Eq.(1) and Eq.(2) we obtain the relationship between the mass fraction of the ejected material ξ and the density ρ as the red curve in Figure 1. My preliminary calculation in Figure 1 suggests that the mixture with $(T_c - T_a) = (T_0 - T_a) < 0.7$ is expected to be buoyant (Figure 1).

Figure 1: The normalized temperature (black curve) and the density (ρ ; red curve) as a function of the mass fraction of ejected materials in the eruption cloud (ξ) based on Eqs. (1) and (2). $T_0 = 1123$ K and $y_w = 0.02$ are assumed. The horizontal thin red line indicates the density of air at $p = 1.013 \times 10^5$ Pa (ρ_{air}).

This simple consideration based on the reference state leads to a conclusion to explain the occurrence of low temperature PDC in Figure 3. The values of T and ξ for $\rho = \rho_{air}$ (the star in Figure 1) primarily depend on T_0 ; they decrease as T_0 becomes low, as is the case of phreatomagmatic eruptions (see Koyaguchi and Suzuki (2018) for more comprehensive discussions). Thus, it is inferred that very low temperature PDCs in Figure 3 resulted from phreatomagmatic eruptions with low T_0 . Figure 2c in this paper, on the other hand, indicates that the variation in PDCs' temperature can result from variation in U_0 (i.e., initial momentum) for given mass eruption rate Q_0 . As U_0 increases for given Q_0 , Q_c decreases (see Appendix below), which generates a PDC with a low $(T_c - T_a) = (T_0 - T_a)$. To apply this result to the observations in nature (e.g., the observations in Figure 3), we must specify the minimum value of $(T_c - T_a) = (T_0 - T_a)$ for the mixture to be denser than air to form a PDC (i.e., whether it is as low as 0.5 like the case of Figure 2c or whether it can be even lower). For this purpose, we must understand the mechanism how the mixtures with $(T_c - T_a) = (T_0 - T_a) < 0.7$ can be denser than air. There may be some possible mechanism to form a dense mixture with low $(T_c - T_a) = (T_0 - T_a)$ as follows.

- The mixture of the erupted material and air is not necessarily a consequence of simple binary mixing, which is assumed in the calculations of Eqs. (1) and (2), but may be heterogeneous. If the thermal equilibrium in the mixture is incomplete, the density of such heterogeneous mixture may deviate from the red curve in Figure 1 to considerable extent (see the light blue region in Figure 1 in this report).
- The above calculations of Eqs. (1) and (2) is based on the assumption that the effect of the particle separation is negligible, whereas Supplementary Figure 3 indicates the effect of the separation of coarse pyroclasts may be significant.

To understand the implication of Figure 2c, we must somehow evaluate the effects of the above mechanisms. Particularly, to clarify the effect of the first mechanism, it is essentially important to describe the heterogeneity of the collapsing down-flow in the simulation results. For example, according to the definition in line 112 of Supplementary information, T_c is calculated as an

averaged temperature; however, it is not clear from the averaged value whether the collapsing down-flow is composed of a uniform mixture with a uniform temperature of T_c or it is composed of heterogeneous mixture with a wide range of T . It is also helpful if we could know the variations in the temperature T , the density ρ and the mass fraction of ejected material ξ at each grid point before averaging; those values of T and ρ can be plotted as a function of ξ in a diagram like Figure 1 in this review report.

I suggest that the geological implication using Figure 3 is possible only after the above discussions on the mixture with low $(T_c - T_a)/(T_0 - T_a)$. Judging from Figure 2c, I have a feeling that the above two mechanisms may explain a dense mixture with $(T_c - T_a)/(T_0 - T_a)$ as low as 0.5; however, to explain the occurrence of PDCs with very low temperature (those of $T = 100\text{--}300$ °C in Figure 3), additional effects such as the presence of ground water would be necessary.

We really appreciate this constructive and well-grounded critique. In response to your comment, we presented a new Figure 2, along with two new Supplementary Figures (Supplementary Figures 2, and 3) and a paragraph to the results section (named: “Thermodynamic constraints for collapsing gas-particle mixtures”) in the revised manuscript that give an explanation about the variations in temperature T , density ρ and mass fraction ξ of the collapsing down-flow mixture, necessary to explain the mechanisms by which a mixture with $(T_c - T_a)/(T_0 - T_a) < 0.7$ can be denser than the surrounding atmosphere. It is revised as follow:

Thermodynamic constraints for collapsing gas-particle mixtures

To better understand the mechanism of temperature drop due to mixing with atmospheric air, we here derive, from thermodynamic considerations, the thermal conditions that would characterize the collapsing mixture. We assume an isobaric transformation between the two following states: initial) the erupted material and air are perfectly separated; final) the two phases are perfectly mixed and the resulting mixture is homogeneous. The final mixture temperature $T(\xi)$ can be expressed, from enthalpy conservation, as a function of the mass fraction of the erupted material ξ ¹³. A dimensionless temperature $\theta(\xi)$ can be written as:

$$\theta(\xi) \stackrel{\text{def}}{=} \frac{T(\xi) - T_{\text{air}}}{T_0 - T_{\text{air}}} = \frac{\xi C_{Pm}}{\xi C_{Pm} + (1 - \xi)C_{P\text{air}}} \quad (1)$$

where T_0 is the initial temperature of the erupted material, T_{air} is the initial air temperature, and $C_{P\text{air}}$ is the specific heat at constant pressure of the air (thermodynamic properties used are listed in Supplementary Table 2). The specific heat at constant pressure of the erupted material is $C_{Pm} = y_w C_{Pw} + (1 - y_w)C_s$, where y_w is the initial gas fraction in the erupted material, C_{Pw} and C_s are the specific heats of the gas and solids, respectively. The ratio between the mixture density $\rho(\xi)$ and the air density ρ_{air} is:

$$\frac{\rho(\xi)}{\rho_{\text{air}}} = \frac{R_{\text{air}} T_{\text{air}}}{[\xi y_w R_w + (1 - \xi)R_{\text{air}}] T(\xi)} \quad (2)$$

where R_w and R_{air} are the gas constant of the erupted gas and air, respectively. Equations (1) and (2) are derived under the following assumptions: 1) the enthalpy is conserved, thus the transformation is at constant pressure and the effects of dissipation and gravity can be disregarded; 2) the mixture in the final state is homogeneous; 3) particle separation can be disregarded; 4) air temperature and density are constant, i.e., atmospheric stratification is neglected; 5) the volume fraction of solid is negligibly small. In the following, we compare theoretical results with the 3D simulation, to evaluate to which extent these assumptions are

correct cell by cell and in the whole collapsing region. From equations (1) and (2), the minimum temperature and mass fraction for which the mixture is non-buoyant ($\rho(\xi) > \rho_{\text{air}}$) can be derived:

$$\theta_{\min} = \frac{\alpha\beta\gamma - 1}{(\alpha - 1)\beta\gamma + \gamma} \quad (3)$$

$$\xi_{\min} = \frac{\beta\gamma - \alpha}{\gamma + 1 - \alpha} \quad (4)$$

$$\alpha = \frac{C_{Pm}}{C_{P\text{air}}}; \quad \beta = \frac{R_{\text{air}}}{R_{\text{air}} - y_w R_w}; \quad \gamma = \frac{T_0}{T_{\text{air}}} - 1.$$

In our case, $\alpha \approx \beta \approx 1$ and equations (1-4) simplify into:

$$\theta(\xi) = \frac{\alpha\xi}{(\alpha - 1)\xi + 1} \approx \xi$$

$$\frac{\rho(\xi)}{\rho_{\text{air}}} = \frac{\beta}{(\beta - \xi)(\gamma\theta + 1)} \approx \frac{1}{(1 - \xi)(\gamma\xi + 1)}$$

$$\theta_{\min} \approx \xi_{\min} \approx \frac{\gamma - 1}{\gamma} = \frac{T_0 - 2 T_{\text{air}}}{T_0 - T_{\text{air}}}$$

As pointed out by ref.¹³, θ_{\min} and ξ_{\min} primarily depend on T_0 and T_{air} . Using $T_0 = 1100$ K and $T_{\text{air}} = 290$ K, we get $\theta_{\min} \approx \xi_{\min} \approx 0.64$. Below this value, a homogeneous mixture that cooled down by the sole effect of air entrainment would necessarily be positively buoyant.

In the volcanic case, stratification would be non-negligible: the air density drop along the collapsing region can be as high as 40% and the temperature drop is about 10% (see Supplementary Fig. 1). In Fig. 2, we show the comparison between the theoretical relationships (1) and (2) (solid curves), and the simulated values of non-dimensional temperature and density in each cell of the collapse region (points). The simulated dimensionless density is evaluated with equation (2), considering atmospheric temperature stratification $T_{\text{air}}(z)$ for each cell in the collapsing region, while its theoretical value is obtained using the average air temperature between the crater and the collapsing height. The two end-member cases, with and without particle decoupling, are compared. In the equilibrium case (dusty-gas; Fig. 2a, c), the thermodynamics of the numerical solution almost perfectly follows the simplified theoretical model in every cell of the collapsing region. This result is two-fold: on one side, it shows that all the hypothesis used to obtain equations (1) and (2) are locally valid; on the other side, it confirms the ability of the numerical solver to accurately solve the enthalpy equation. In the case of particle decoupling (Fig. 2b, d), both temperature and density values deviate from equations (1) and (2), although following the expected theoretical trend. Since in this case we cannot use equations (1) and (2) to define θ_{\min} and ξ_{\min} , these parameters are evaluated by searching the last point with negative buoyancy (Fig. 2b, d). Particle separation makes θ_{\min} and ξ_{\min} smaller than those predicted by equations (3) and (4). Consequently, particle clustering and settling enhance Q_c and the temperature drop. For the two end-members, we find $Q_c = 15\%$ and $Q_c = 79\%$ with particle decoupling, while $Q_c = 4\%$ and $Q_c = 70\%$ without. We quantify this effect in Supplementary Fig. 3, where a larger temperature drop is observed because of particle decoupling (up to 20% for the partial collapse end-member).

The distribution of the points along the whole theoretical curves of Fig. 2 clearly indicates that the collapsing region is highly heterogeneous, both in terms of particle concentration and temperature. Thus, even if equations (1) and (2) hold locally, they cannot be used to estimate the global temperature and buoyancy of the collapsing region. To estimate the average temperature T_c in the heterogeneous collapse region, we used instead a weighted average (described in the Methods section), based on the mass distribution in every cell. Its non-dimensional value is reported in Figure 2 (red line) as θ_c . In the partial collapse case (Fig. 2a, b), θ_c is smaller than θ_{\min} , due to the heterogeneity and enhanced entrainment in the

collapsing stream. In the fountaining regime (Fig. 2c, d) $\theta_c > \theta_{\min}$, clearly indicating the establishment of full collapsing conditions with a smaller entrainment efficiency.

Despite such complexities in the collapsing region, T_c (or θ_c) results to be a good estimator for the average basal temperatures of PDCs in the two end-member simulations. As shown in Supplementary Fig. 4, for the partial collapse case, $T_c = 790$ K and the average PDC temperature $T_{PDC} = 809$ K, while for the total collapse end-member $T_c = 1090$ K and $T_{PDC} = 1062$ K. The good agreement between these estimates confirms the validity of our approach.

Figure 2. Thermodynamic properties of the collapse region for the reference models. Dimensionless temperature θ and density ρ / ρ_a as a function of erupted material mass fraction ξ in the collapsing region. Blue and red points are extracted from the numerical simulation data (temperature, mass fractions, density) and the atmospheric temperature in each cell of the collapse region (see Supplementary Fig. 1). Yellow and green solid lines are obtained from equations (1) and (2), using an average atmospheric temperature

between the crater and the collapse height. The red solid line highlights the non-dimensional value of the average temperature of the collapsing region θ_c (computed as described in the Methods section), while the black solid line is the level of neutral buoyancy. Black dotted and dashed lines indicate θ_{\min} and ξ_{\min} , respectively. All the blue points below θ_{\min} and all the red points left of ξ_{\min} are less dense than the atmosphere at the same height. Panels **a** and **b** are for the partial collapse end-member, while **c** and **d** are for the total collapse. Panels **a** and **c** are for the dusty-gas model, while **b** and **d** take into account particle decoupling.

Supplementary Figure 3 | Comparison between dusty-gas and kinematic decoupling simulations.

Relationship between the percentage of collapsing mass Q_c and its corresponding average temperature value T_c normalized to the initial magmatic temperature T_0 with respect to the atmospheric air temperature T_a , for dusty-gas (black symbols) and kinematic decoupling (red symbols) simulations with the same initial eruptive conditions (mass eruption rate $Q = 10^8$ kg/s; temperature $T = 1123$ K; water content $0.5 < y_w < 2.5$ wt%; pressure $0.1 < P < 10$ MPa). The black and red solid lines show the linear least square regression for dusty-gas and kinematic decoupling simulations, respectively. The red regression line is drawn using all data from kinematic decoupling simulations (it is the same shown in Figure 3c).

Supplementary Figure 4 | Time series of PDC temperature data for the reference models. Temporal evolution of the mixture temperature (black solid line) in the bottom layer of PDC generated by the partial (a) and total (b) column collapse end-members. Temperature is measured at a distance from the injection point of 10 inlet radii ($r = 10r_0$). Green solid lines indicate the time-averaged temperature of each pulse. The PDC temperatures (T_{pdc}) averaged over the reported time-window are 809 K and 1062 K for the partial and total collapse end-members, respectively. Such temperatures are very similar (within less than 5% in this case) to the mean temperatures at collapse.

To sum up, Reviewer #2 pointed out two possible phenomena to explain a collapsing mass with $(T_c - T_a)/(T_0 - T_a) < 0.7$: 1) heterogeneity; 2) gas-particle disequilibrium.

The second mechanism contributes to the local (i.e. cell-by-cell) deviation from the theoretical law, and we were able to quantify such an effect (Figs. 2b,d). In the manuscript, we also listed: gravity effects, dissipation, atmospheric stratification, volume of the solid, as other assumption needed to get the Eqs. (1) and (2) suggested by the Reviewer. Moreover, we modify the theoretical equation for the density writing directly the density ratio $\frac{\rho(\xi)}{\rho_{air}}$ to reduce the effects due to atmospheric stratification.

However, the main source of discrepancy between the predicted minimum temperature that a homogeneous collapsing column can reach, and the calculated averaged temperature is due to the heterogeneity of the collapsing region. Since the collapsing region is highly heterogeneous one (as shown by the new Figure 2), the collapsing temperature needs to be defined by some averaging procedure. We decided to weight-average the temperature with the mass contained in each cell. As a further check of the robustness of the procedure, we compared the mean temperature T_c in the collapsing region with the temperature at the base of the forming PDCs T_{pdc} (Supplementary Fig. 4). It turns out that $T_c \sim T_{pdc}$ for both the end-member cases. The good agreement between these estimates confirms the validity of our approach. In Fig. 2 we also quantify the effect of particle decoupling on the conservation of the enthalpy. Decoupling enhances column collapse.

We believe that the new analytical and numerical analysis proposed by Reviewer #2 strengthens the understanding of the emplacement temperature variability of pyroclastic flow deposits shown in Figure 4 (which, as suggested by Reviewer #1, we amended to include the percentage of collapsing mass derived by our models, hoping that this addition makes comparison between the observed temperature data and our model results more quantitative).

We would like to thank you again for allow us to verify that the ASHEE code properly solve the energy balance equation.

Specific comments:

- (1) Lines 62–67: From the description of *Supplementary Methods* I could infer that the 3-D numerical simulations in this study are actually those for pressure-balanced jets with variable U_0 (i.e., so-called correctly expanded jets); however, the main text (as well as Table 1) gives an impression that the simulations are done for variable U_v and P_v , which is misleading. In this study, the physical quantities at $P = P_0$ (atmospheric pressure) are assumed to be uniform (i.e., top-hat distributions). On the other hand, recent numerical studies (Carcano et al., 2013; Koyaguchi et al., 2018) indicate that, when the jet is not pressure-balanced at the vent, the decompression and compression processes just above the vent strongly modify the distributions of the physical quantities at $P = P_0$. As is discussed in *Major comment*, the distributions of the physical quantities in the column (e.g., the heterogeneity of T) is considered to be critical to generate a dense mixture with a relatively low $(T_c - T_a) = (T_0 - T_a)$. To avoid readers' misunderstanding, the author should explicitly state in the main text that the 3-D numerical simulations in this study are essentially those for pressure-balanced jet with variable U_0 , and, if possible, discuss how the assumption of top-hat distributions of physical quantities at $P = P_0$ affects the main conclusions of this paper.

We confirm (and we have better stated in the text at Lines 78–80, manuscript with highlighted changes) that source conditions for simulations are set for a pressure-balanced (correctly-expanded) jet with a *top-hat* profile (i.e., uniform inlet conditions). The choice of limiting to these conditions was motivated: 1) to avoid simulating the decompression above the conduit exit, requiring very high-resolution grids at the inlet (that would have needed an enormous computational effort) and 2) to simplify the source parameters when focusing on the collapse

dynamics. We are aware, especially after reading the more recent Koyaguchi and Suzuki 2018 paper, of the potential impact that a more accurate description of the compression/decompression patterns might have on the collapse dynamics. We have therefore welcomed Reviewer #2 suggestion to comment about this issue (Lines 81–86, manuscript with highlighted changes), which will be the topic for a future work.

We also agree with Reviewer #2, that we might have imposed directly the mixture parameters at the inlet, instead of imposing them at the conduit exit and computing analytically the conditions after the *correct expansion*, and that this can be somehow misleading for the reader. However, not only the two choices are completely equivalent, but we believe that ours allows a much direct linking to vent conditions deriving from conduit flow models. This is the approach that we followed also in our previous works (e.g., Esposti Ongaro et al., 2008).

We finally remark that the term *vent* in our paper refers to the conduit exit, not the crater exit. There is unfortunately not a general consensus in the literature about such nomenclature, and we preferred to be consistent with our previous works.

Esposti Ongaro, T. et al. Transient 3D numerical simulations of column collapse and pyroclastic density current scenarios at Vesuvius. J. Volcanol. Geotherm. Res. 178, 378–396 (2008).

- (2) Lines 95–97: The definition of partial collapse based on Q_c is rather *indirect* and it requires more explanations. I agree that Q_c represent something important for understanding of partial collapse, but I am not fully convinced that Q_c represents the fraction of collapsing mass as a PDC. There are at least two ambiguities. The separation of particles results in mass loss from an ascending column. Therefore, strictly speaking, the difference between the mass flow rate at the black dot and the total mass eruption rate does not represent the collapsing mass flow rate as a PDC. Considering the presence of large scale eddy around the top of fountain (H_c), a part of erupted mass may also be trapped by the large scale circulation even though the whole down-flow is not denser than the ambient air. These effects of the particle separations and circulations should be quantitatively evaluated to understand the meaning of Q_c . In my opinion, to clarify whether Q_c definitely represents the fraction of collapsing mass as a PDC, it is essentially important to show that the collapsing down flow is denser than the ambient air.

We thank Reviewer #2 for this comment. As suggested, we now explain in more detail our motivations to choose Q_c as a measure of the collapsing intensity. In the Methods section, we also quantify the effect of the recirculation eddy (lines 430-451 in the manuscript with highlighted changes). It is instead very difficult to directly quantify the amount of mass going into the PDC, because while the mixture spreads horizontally it continuously loses the mass that becomes positively buoyant, as a result of the entrainment and mixing with the atmosphere. In other words, where is the edge between the collapsing region and the beginning of the PDC?

For this reason, we decided to use a clearly defined method where the collapsing mass is obtained by comparing the mass flow rate of the ascending column above the collapsing region with the mass eruption rate. Thanks to conservation of mass, all the mass not present in the ascending column should have been lost by the collapsing region and the recirculation eddy.

As we describe in our previous reply to the Reviewer's major comment, the collapsing region is heterogeneous, so it contains both positively and negatively buoyant parcels (cells).

- (3) Line 103: What is the definition of H_{jet} ? Does it the level of the minimum point of horizontally integrated velocity in Figure 1? If so the level does not necessarily represent the level where the vertical momentum of the whole eruptive column is exhausted. The profile of horizontally integrated velocity is determined by competing effects of decreasing momentum due to negative buoyancy and increasing momentum of positive buoyancy. If the effect of increasing buoyancy due to positive buoyancy is large, the upward velocity may have a minimum point at a lower level before the initial momentum is exhausted.

Thank you for highlighting this point. We have clarified the explanation of our H_{jet} measurement strategy further, as we realized how this can be misleading. As we now specify in the manuscript (lines 143-146/419-429 in the manuscript with highlighted changes), we decided to evaluate H_{jet} by using a method that is not based on any assumption on the entrainment profile, but it is only based on the velocity profile. The other two methods that we now refer to in the paper, the Morton length scale generalized to the non-Boussinesq case and the method defined by Koyaguchi and Suzuki (2018), require the definition of the entrainment coefficient (to be either 0 or a constant value). In any case, we show that our methodology to evaluate H_{jet} gives broadly comparable results to those obtained with the other two methods.

Koyaguchi, T. & Suzuki, Y. J. The Condition of Eruption Column Collapse: 1. A Reference Model Based on Analytical Solutions. J. Geophys. Res. Solid Earth 7461–7482 (2018). doi:10.1029/2017JB015308

- (4) The result in Figure 2b (and related text from line 141 to 145) is interesting, although it may not be the main subject of this paper. In fact, similar tendencies have been pointed out in Costa et al. (2018) and Koyaguchi et al. (2018). Although the present result in Figure 2b is more comprehensive and looks more systematic, those previous works can be cited here.

We have now expanded the discussion around Figure 2b (please refer to lines 264-276/332-335 in the manuscript with highlighted changes), referring to the works suggested by the Reviewer. Based on Figure 2b, we also emphasize the importance of not only considering the effects of wind but also the collapse regime for determining the total plume height, which, for a given mass eruption rate, can be substantially reduced in the case of a partial collapse.

- (5) The caption of Supplementary Figure 3 needs to be corrected. There seem to be two important typos. Judging from the value of H_c (i.e., the positions of the black dots), the black curves are the results of partial collapse and the pink curves are the results of the near total collapse. In the pink curves, the dotted curve shows nearly zero mass flow rate at high altitudes. According to the caption, the upper part of the this column (with pink color) is composed of coarse particles shown by dashed curve alone and almost fully

depleted in fine particles, which does not make sense to me. I suspect that dashed and dotted curves represent the mass flow rate of fine and coarse particles, respectively.

Many apologies – Yes, the Reviewer is right. Thanks so much for catching this misstatement about the dashed and dotted curves that have now been corrected.

Minor points annotated in the manuscript:

Line 39 – The initial momentum (U_0) is the other critical factor (e.g., Koyaguchi and Suzuki, 2018). See also the appendix of review report.

We have now added the reference suggested by the Reviewer (Lines 42-44, manuscript with highlighted changes).

Line 103 – How did you determined H_{jet} ?

See comment (3).

Line 104 – I could not follow the logic of this part.

We rephrased this sentence in the revised manuscript (Lines 147-151 in the manuscript with highlighted changes) to remove the ambiguity. What we are saying is that when the jet becomes gradually more dilute due to turbulent shearing at its margins, then $H_{un} \gg H_{jet}$, since the jet core remains unmixed. This is in accordance with what was previously suggested by Suzuki and Koyaguchi (2012) and Koyaguchi and Suzuki (2018).

Suzuki, Y. J. & Koyaguchi, T. 3-D numerical simulations of eruption column collapse: Effects of vent size on pressure-balanced jet/plumes. J. Volcanol. Geotherm. Res. 221–222, 1–13 (2012).

Koyaguchi, T. & Suzuki, Y. J. The Condition of Eruption Column Collapse: 1. A Reference Model Based on Analytical Solutions. J. Geophys. Res. Solid Earth 7461–7482 (2018). doi:10.1029/2017JB015308

Line 137 – Is this idea still widely accepted? I think that since Dade and Huppert (1996) and Bursik and Woods (1996) the runout distance of Taupo has been interpreted in a different way.

We thank the Reviewer for pointing our attention to the work of Dade and Huppert (1996) and Bursik and Woods (1996), to which we now refer in the manuscript. Their works introduced the concept that implicitly and qualitatively is similar to ours, but no one have explicitly discounted the idea that very high collapse heights generate faster PDCs with longer runouts. Although such concept may seem obvious for some, there is still a misconception in modern Volcanology that the higher the collapse height the faster the PDCs velocity and the longer the runout distance (e.g., Cole, P.D., Neri, A., & Baxter, P.J., 2015). For example, the Energy Cone Model is based on the assumption that the PDCs' velocity and runout are related with the initial potential energy, and so with the initial vertical position of the mass that will form the PDCs. Such model is still largely used in the volcanological community (e.g., Tierz et al., 2016). However, our data show

that the effect of the collapse regime can be dominant and so it should be carefully taken into account in future studies that assess PDCs' mobility. As also suggested by Reviewer #1, we reworded a bit the discussion on this point (see Lines 260-263 in the manuscript with highlighted changes) to make it more clear.

Bursik, M. I. & Woods, A. W. The dynamics and thermodynamics of large ash flows. Bull. Volcanol. 58, 175–193 (1996).

Cole, P. D., Neri, A. & Baxter, P. J. in The Encyclopedia of Volcanoes 943–956 (Elsevier, 2015).

Dade, W. B. & Huppert, H. E. Emplacement of the Taupo ignimbrite by a dilute turbulent flow. Nature 381, 509–512 (1996).

Tierz, P. et al. Suitability of energy cone for probabilistic volcanic hazard assessment: validation tests at Somma-Vesuvius and Campi Flegrei (Italy). Bull. Volcanol. 78, (2016).

Points annotated in the Supplementary Material (now Methods section)

Line 25 – This does not necessarily mean the maximum exit velocity. Correctly-expanded flow can have a larger exit velocity (see Koyaguchi et al. 2010).

This wasn't well explained in the previous version of the manuscript. Choked flow (sonic) conditions represent the maximum velocity at the conduit exit. We agree that the mixture can accelerate when it expands. We have simply deleted this part from the text, as not necessary (Lines 72-77/376-378, manuscript with highlighted changes).

Line 37 – You are using only U_0 as a boundary condition for the 3D simulations, aren't you? In other words, the 3D simulations are done for pressure-balanced jets (correctly expanded supersonic (or sonic) flow with variable Mach number) with top-hat distribution (see comment for Supplementary Figure 1, too). Why is the calculation using U_v and P_v necessary? Considering that the distributions of physical quantities for free decompression flow are far from top-hat distributions, it is less misleading to define the vent condition using U_0 rather than U_v and P_v . See Specific comment #1.

See Response to Specific comment #1.

We agree with Reviewer #2, that we might have imposed directly the mixture parameters at the inlet, instead of imposing them at the conduit exit and computing analytically the conditions after the *correct expansion*, and that this might be somehow misleading for the reader. However, not only the two choices are completely equivalent, but we believe that ours allows a much direct linking to vent conditions deriving from conduit flow models. This is the approach that we have followed also in our previous works (e.g., Esposti Ongaro et al., 2008).

Esposti Ongaro, T. et al. Transient 3D numerical simulations of column collapse and pyroclastic density current scenarios at Vesuvius. J. Volcanol. Geotherm. Res. 178, 378–396 (2008).

Line 48 – Are those quantities assumed to be uniform at the level of $P_0=P_{atm}$ (i.e., top hat distribution)? If so the distributions of those quantities are quite different from those for free decompression (i.e., $U_v=C$ and $P_v>P_0$) flow after the decompression process (Ogden et al., 2008; Carcano et al., 2013; Koyaguchi et al., 2018).

See Response to Specific comment #1.

We confirm that the flow quantities are uniform at the level of $P_0=P_{atm}$. We agree that these conditions are not achieved in a free decompression but for a correctly-expanded flow, and we have specified that at Lines 78-86 in the manuscript with highlighted changes.

Line 69 – According to Koyaguchi and Suzuki (2018), this curve (Eq. (2)) can be approximated by a simple equation $U_0/U_v=2-P_0/P_v$ for free decompression flow like the present cases. So the figure like this may not be necessary.

See Response to Specific comment #1.

We agree and thank the Reviewer for the suggestion. We have eliminated the Figure and reported the asymptotic behavior of Eq. 2 (now Eq. 6) at Lines 406-410 in the manuscript with highlighted changes.

Line 91 – See Specific comment #3.

We think the Reviewer is referring to Specific comment #2 here (about Q_c calculation). Thus, see response to Specific comment #2

Line 110 – The weight flux average may be better than the weight average, although it is a rather difficult problem.

Here we wanted to estimate the average temperature of the collapsing region, not its dynamical properties. The weight flux would have given us information about the energy flux in the collapsing region, not just about its temperature.

Appendix: Consistency and discrepancy with the column collapse condition proposed by Koyaguchi and Suzuki (2018) and Koyaguchi et al. (2018)

In the course of review process, I have confirmed how the present numerical results are consistent with the theoretical and numerical column collapse condition by Koyaguchi and Suzuki (2018) and Koyaguchi et al. (2018). Koyaguchi and Suzuki (2018) have shown that the column collapse condition of a pressure-balanced jet (e.g., Bursik and Woods, 1991) is expressed by a simple relationship as

$$M_a/\tilde{Q}^{1/5} = 1 \quad (3)$$

where M_a is the Mach number after decompression ($U_0/\sqrt{\gamma_w R_w T_0}$ using the present notation) and \tilde{Q} is the dimensionless mass eruption rate (see Koyaguchi and Suzuki (2018) for the normalization factor). An eruption column is stable to form a buoyant plume when $M_a/\tilde{Q}^{1/5} > 1$, whereas it collapses to form PDCs as $M_a/\tilde{Q}^{1/5} < 1$. Using the values in Table 1, we can plot Q_c (%) as a function of $M_a/\tilde{Q}^{1/5}$ for the present numerical results (Figure 2). The present results are consistent with the prediction by Koyaguchi and Suzuki (2018) in the sense that $Q_c = 0$ (i.e., fully buoyant) for $M_a/\tilde{Q}^{1/5} > 1$. The present results also show that Q_c gradually increases with decreasing $M_a/\tilde{Q}^{1/5}$ in the region of $M_a/\tilde{Q}^{1/5} < 1$. It is interesting to see the variation of Q_c with $M_a/\tilde{Q}^{1/5}$ collapses to a single relationship for $Q_0 = 10^7$ – 10^9 kg/s; this may suggest that the present numerical study has captured a novel universal feature of the transition from a buoyant plume to total collapse.

Figure 2: Q_c (%) as a function of $M_a/\tilde{Q}^{1/5}$. For the definition of M_a and $\tilde{Q}^{1/5}$ see text. The color of symbols are the same as the manuscript. Green: $Q_0 = 10^7$ kg/s, red: 10^8 kg/s, and black: 10^9 kg/s.

It should be noted that the present numerical results and those of Koyaguchi et al. (2018) do not perfectly coincide with each other in a quantitative sense. The transitional state (i.e., partial collapse etc) is observed in the range of $M_a/\tilde{Q}^{1/5} < 1$ in the present study. On the other hand, it is typically observed in the range $1 < M_a/\tilde{Q}^{1/5} < 1.2$ in the 3-D simulations of Koyaguchi et al. (2018); the range of the transitional zone varies with Q_0 and P_v to a considerable extent in their simulations. This quantitative discrepancy may be caused by the fact that the present simulations take the effects of particle settling into account, whereas those of Koyaguchi et al. (2018) do not. The discrepancy may partly result from the difference in the condition at the level of $P = P_0$. In Koyaguchi et al. (2018), the boundary condition of $P_v \neq P_0$ is set at the crater top such that the physical quantities at the level of $P = P_0$ is heterogeneous because of the decompression/compression processes; this heterogeneity causes the diverse features in the transitional state in their simulations. In this study, on the other hand, a top hat distribution is imposed at the level of $P = P_0$. It is suggested that further study would be necessary to fully understand the diverse features in the transitional state.

We are very pleased to verify that our collapsing conditions fit with those predicted by Koyaguchi and Suzuki 2018, and Koyaguchi et al., 2018, as shown in Fig 2 of Reviewer's comments. As Reviewer #2 noted in his last sentence, further studies would be needed to fully investigate the transitional regimes in comparison with predictions proposed in Koyaguchi et al., 2018. Overall, while we do think this is a really interesting area for further extension, we would like to point out that this kind of analysis does not affect the quality and meaning of our results, and addressing this here is beyond the scope of the present study. We aim to further investigate these aspects in a forthcoming paper.

Koyaguchi, T. & Suzuki, Y. J. The Condition of Eruption Column Collapse: 1. A Reference Model Based on Analytical Solutions. J. Geophys. Res. Solid Earth 7461–7482 (2018). doi:10.1029/2017JB015308

Koyaguchi, T., Suzuki, Y. J., Takeda, K. & Inagawa, S. The condition of eruption column collapse: Part 2. Three-dimensional (3D) numerical simulations of eruption column dynamics. J. Geophys. Res. Solid Earth (2018).

REVIEWERS' COMMENTS:

Reviewer #1 (Remarks to the Author):

Looks great! Good job!

Reviewer #2 (Remarks to the Author):

The authors made an excellent reply to all my comments.

The new analyses shown in new Figure 2 have greatly improved our understanding of this subject.

The discussion concerning geological implication based on new Figure 4c is convincing now.

Takehiro Koyaguchi